# NEXTSTEP-1: TOWARD AUTOREGRESSIVE IMAGE GENERATION WITH CONTINUOUS TOKENS AT SCALE

**Chunrui Han**[1,†]   **Guopeng Li**[1,†]   **Jingwei Wu**[1,†]   **Yan Cai**[1,3,†,⋆]   **Yuang Peng**[1,2,†]
**Deyu Zhou**[1,⋆]   **Haomiao Tang**[1,2,⋆]   **Hongyu Zhou**[1]   **Kenkun Liu**[1,⋆]   **Binxing Jiao**[1]
**Daxin Jiang**[1]   **Xiangyu Zhang**[1]   **Yibo Zhu**[1]   **Zheng Ge**[1]   **Shu-Tao Xia**[2,¶]   **Quan Sun**[1,¶]

[1]StepFun   [2]Tsinghua University   [3]Peking University

 GitHub   🤗 HuggingFace   NextStep-1   NextStep-1.1

## ABSTRACT

Prevailing autoregressive (AR) models for text-to-image generation either rely on heavy, computationally-intensive diffusion models to process continuous image tokens, or employ vector quantization (VQ) to obtain discrete tokens with quantization loss. In this paper, we push the autoregressive paradigm forward with **NextStep-1**, a 14B autoregressive model paired with a 157M flow matching head, trained on discrete text tokens and continuous image tokens with next-token prediction objectives. NextStep-1 achieves state-of-the-art performance for autoregressive models in text-to-image generation tasks, exhibiting strong capabilities in high-fidelity image synthesis. Furthermore, our method shows strong performance in image editing, highlighting the power and versatility of our unified approach. To facilitate open research, we have released our code and models to the community at `https://github.com/stepfun-ai/NextStep-1`.

## 1 INTRODUCTION

Diffusion models have firmly established themselves as the dominant paradigm for high-fidelity image synthesis (Podell et al., 2024; Esser et al., 2024; Labs, 2024; Wu et al., 2025a). Despite this success, recent state-of-the-art architectures typically operate in a decoupled manner, relying on separate, pre-trained text encoders (e.g., T5 (Raffel et al., 2020) or CLIP (Radford et al., 2021)) to feed semantic features into a Multimodal Diffusion Transformer (MMDiT) (Esser et al., 2024) backbone via cross-attention. This non-end-to-end design imposes rigid constraints: it fixes the input context window, restricts deep multimodal fusion, and hinders the model's ability to handle arbitrary interleaved image-text sequences naturally.

Inspired by the scalability of Large Language Models (LLMs)(OpenAI, 2025a), the "next-token prediction" paradigm offers a compelling alternative for unified multimodal generation. Existing approaches generally fall into two categories, each with distinct limitations. The first category, represented by discrete autoregressive models(Sun et al., 2024a; Chen et al., 2025b; Wang et al., 2024b), relies on Vector Quantization (VQ)(Zheng et al., 2022; Eslami et al., 2021) to discretize images into visual tokens. This process introduces an information bottleneck—manifesting as reconstruction artifacts—and suffers from exposure bias(Han et al., 2025). Furthermore, to mitigate information loss, models like Emu3 (Wang et al., 2024b) must employ low compression rates, resulting in excessively long token sequences that inflate computational costs and training difficulty. The second category involves hybrid architectures, such as Transfusion (Zhou et al., 2025) and Bagel (Deng et al., 2025), which attempt to integrate noisy inputs and diffusion losses directly into the LLM. While promising, these methods often compromise training efficiency. Unlike pure next-token prediction, they typically require processing both noisy latents and clean conditioning signals (or ground truth images) simultaneously within bidirectional attention blocks. This data duplication significantly increases the sequence length and attention overhead, undermining the efficiency benefits traditionally associated with sparse autoregressive modeling. Besides, while recent explorations into continuous latent

---

[†]Equal contribution (alphabetical by first name). [⋆]Internship at StepFun. [¶]Corresponding authors.

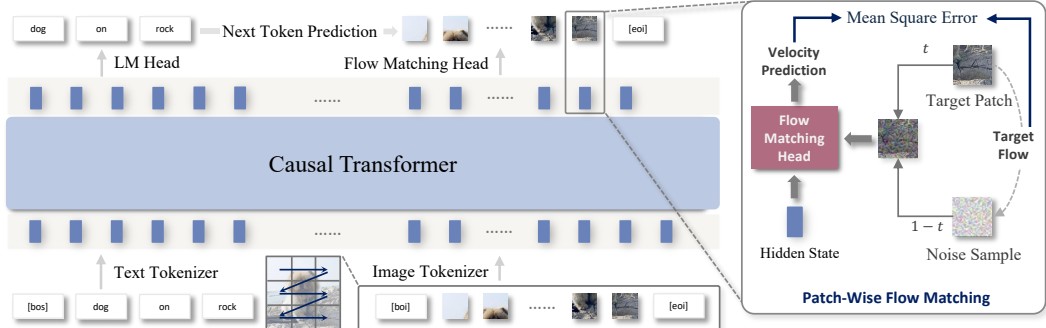

Figure 1: Overview of NextStep-1 Framework. NextStep-1 employs a causal transformer to process tokenized text and image tokens. During training, Flow Matching Head predicts the continuous flow from a noise sample to the next target image patch, conditioned on the output hidden state. At inference, this allows for generating images by iteratively guiding noise to create the next patch.

autoregression (Li et al., 2024c; Fan et al., 2024) attempt to bypass quantization, a substantial performance gap persists. To date, no continuous autoregressive model has matched the visual quality and consistency of state-of-the-art diffusion models while retaining the simplicity and scalability of a standard LLM.

In this work, we demonstrate that the key to closing the performance gap lies in the image representation. We introduce NextStep-1, an autoregressive model built around a novel image tokenizer that is specifically designed to create a well-dispersed and normalized latent space, thereby enabling stable training with high-dimensional continuous latents. By addressing the challenges in image tokenization, we demonstrate that a simple causal transformer with a lightweight flow matching head can achieve state-of-the-art results that rival top diffusion models in both quality and prompt adherence. We showcase the generation quality of our model in Section A. Our contributions are highlighted as follows:

1. We present NextStep-1, an autoregressive text-to-image model that achieves state-of-the-art performance with minimalist architecture, outperforming previous autoregressive methods as well as many strong diffusion-based methods.

2. We reveal key design principles for training a robust image tokenizer that enables stable autoregressive generation with high-dimensional continuous latents.

3. We demonstrate the state-of-the-art performance and versatility of NextStep-1 through comprehensive evaluations on benchmarks for both text-to-image generation (e.g., WISE, GenAI-Bench) and instruction-based image editing (e.g., GEdit-Bench), highlighting its superior capabilities in compositional reasoning and prompt fidelity.

## 2 METHOD

### 2.1 UNIFIED MULTI-MODEL GENERATION WITH CONTINUOUS VISUAL TOKENS

NextStep-1 extends the well-established autoregressive language modeling paradigm to image generation through a simple and intuitive architecture, as illustrated in Figure 1. To unify multi-modal inputs into a single sequence, the images will be tokenized to **continuous image tokens** by the image tokenizer and combined with discrete text tokens. For a multimodal token sequence $x = \{x_0, x_1, ..., x_n\}$, where $x_i$ is either a discrete text token or a continuous visual token, the autoregressive objective under the unified sequence is formalized as:

$$p(x) = \prod_{i=1}^{n} p(x_i \mid x_{<i}).$$
(1)

The unified multi-modal generation task proceeds by sampling the next token $x_i$ from the conditional distribution $p(x_i \mid x_{<i})$ modeled by a network. Discrete text tokens are sampled via a language modeling head, while continuous image tokens are sampled by a flow-matching head.

Our training objective consists of two distinct losses: a standard cross-entropy loss for discrete text tokens, and a flow matching loss (Lipman et al., 2023b) for continuous image tokens. Specifically, the flow matching loss is the mean squared error between the predicted and target velocity vectors that map a noised patch to its corresponding clean patch. The model is trained end-to-end by optimizing a weighted sum of these two losses:

$$\mathcal{L}_{\text{total}} = \lambda_{\text{text}}\mathcal{L}_{\text{text}} + \lambda_{\text{visual}}\mathcal{L}_{\text{visual}} \tag{2}$$

where $\mathcal{L}_{\text{text}}$ and $\mathcal{L}_{\text{visual}}$ denote the loss for text and image tokens respectively, which are balanced by the hyperparameters $\lambda_{\text{text}}$ and $\lambda_{\text{visual}}$.

## 2.2 Model Architecture

**Image Tokenizer.** Our image tokenizer is fine-tuned from flux VAE (Labs, 2024) with only reconstruction and perceptual losses. The tokenizer first encodes an image into 16-channel latents $z$, applying an $8\times$ spatial downsampling factor. To stabilize and normalize the latent space, we apply token-wise normalization (detailed in Appendix C), standardizing each channel to zero mean and unit variance. Furthermore, to enhance the robustness of the image tokenizer and encourage a more uniform latent distribution, we introduce a stochastic perturbation to the normalized latents. This technique is adapted from $\sigma$-VAE (Sun et al., 2024c), where it was employed to prevent variance collapse.

$$\tilde{z} = \text{Normalization}(z) + \alpha \cdot \varepsilon, \quad \text{where } \alpha \sim \mathcal{U}[0, \gamma] \text{ and } \varepsilon \sim \mathcal{N}(0, I) \tag{3}$$

where $\varepsilon$ is standard Gaussian noise, and its magnitude is scaled by a random factor $\alpha$ sampled uniformly from $[0, \gamma]$. The $\gamma$ is a hyperparameter controlling the maximum noise intensity.

The latents from the image tokenizer are pixel-shuffled into a more compact sequence. This is achieved by applying a space-to-depth transformation with a $2\times2$ kernel, which flattens $2\times2$ spatial latents into the channel dimension. For example, this converts the latents of a $256\times256$ image into $16\times16$ grid of 64-channel tokens. This grid is then flattened into a 1D sequence of 256 tokens to serve as input for the following Causal Transformer.

**Causal Transformer.** We initialize our model from the decoder-only Qwen2.5-14B (Yang et al., 2024), leveraging its strong language understanding and reasoning capabilities for text-to-image generation. We organize the multimodal input sequence in the following format:

$$\{\text{text}\} <\text{image\_area}>\text{h*w} <\text{boi}>\{\text{image}\} <\text{eoi}>...$$

where {text} denotes discrete text tokens, and {image} represents continuous image tokens. <boi> and <eoi> are special tokens marking the beginning-of-image and end-of-image. <image\_area>h*w represents the metadata about the spatial dimensions of the 2D image tokens.

Then the output hidden states from LLM are passed to two lightweight heads for modality-specific loss:

- **Language Modeling Head.** We compute Cross-Entropy loss for hidden states of texts.
- **Patch-wise Flow Matching Head.** Following (Li et al., 2024c), we use each patch-wise image hidden states as condition, denoise target patch at timesteps t, and compute the patch-wise flow-matching loss (Lipman et al., 2023a) with a 157M, 12-layer, and 1536 hidden-dimensions MLP.

For positional information, we use the standard 1D RoPE (Su et al., 2024). Despite the availability of more complex 2D or multimodal RoPE alternatives (Bai et al., 2025; Wang et al., 2024a), we found that the simple 1D formulation remains highly effective for mixed text-image sequences, and thus retain it for simplicity and efficiency.

## 3 Training Recipes

### 3.1 Training Image Tokenizer

Our image tokenizer is initialized from the Flux.1-dev VAE (Labs, 2024), selected for its strong reconstruction performance. We fine-tune this model on the image-text dataset detailed in Section B.2 to adapt it to our specific data distribution. For optimization, we employ the AdamW

Table 1: Training recipe of NextStep-1.

| | Pre-Training | | | Post-Training | |
|---|---|---|---|---|---|
| | **Stage1** | **Stage2** | **Annealing** | **SFT** | **DPO** |
| **Hyperparameters** | | | | | |
| Learning Rate (Min, Max) | $1 \times 10^{-4}$ | $1 \times 10^{-5}$ | $(0, 1 \times 10^{-5})$ | $(0, 1 \times 10^{-5})$ | $2 \times 10^{-6}$ |
| LR Scheduler | Constant | Constant | Cosine | Cosine | Constant |
| Weight Decay | 0.1 | 0.1 | 0.1 | 0.1 | 0.1 |
| Loss Weight (CE : MSE) | (0.01 : 1) | (0.01 : 1) | (0.01 : 1) | (0.01 : 1) | - |
| Training Steps | 200K | 100K | 20K | 10K | 300 |
| Warm-up Steps | 5K | 5K | 0 | 500 | 200 |
| Sequence Length per Rank | 16K | 16K | 16K | 8K | - |
| Image Area (Min, Max) | 256×256 | (256×256, 512×512) | (256×256, 512×512) | (256×256, 512×512) | (256×256, 512×512) |
| Image Tokens (Min, Max) | 256 | (256, 1024) | (256, 1024) | (256, 1024) | (256, 1024) |
| Training Tokens | 1.23T | 0.61T | 40B | 5B | - |
| **Data Ratio** | | | | | |
| Text-only Corpus | 0.2 | 0.2 | 0.2 | 0 | - |
| Image-Text Pair Data | 0.6 | 0.6 | 0.6 | 0.9 | - |
| Image-to-Image Data | 0.0 | 0.0 | 0.1 | 0.1 | - |
| Interleaved Data | 0.2 | 0.2 | 0.1 | 0 | - |

optimizer (Loshchilov and Hutter, 2019) with $(\beta_1 = 0.9, \beta_2 = 0.95, \varepsilon = 1 \times 10^{-8})$ for its convergence stability. The model is trained for 50K steps with a total batch size of 512, using a constant learning rate of $1 \times 10^{-5}$ preceded by linear warm-up of 1000 steps.

## 3.2 PRE-TRAINING

The specific hyperparameters and data ratios for our pre-training are detailed in Table 1. Specifically, the pre-training follows a three-stage curriculum designed to progressively refine the model's capabilities. Throughout these stages, all model parameters are trained end-to-end except for the pre-trained image tokenizer.

**Stage1.** In this initial stage, the model learns a foundational understanding of image structure and composition. For computational efficiency, all images are resized and randomly cropped to a fixed 256×256 resolution. The training curriculum is composed of a diverse data mixture: 20% text-only corpora, 60% image-text pairs, and 20% interleaved data. This stage consumed approximately 1.23T tokens.

**Stage2.** We implement a dynamic resolution strategy to train the model on a range of higher resolutions, targeting 256×256 and 512×512 base areas. This strategy utilizes different aspect ratio buckets for computational efficiency. In this stage, we enrich the data mixture with more text-rich and video-interleaved data, leveraging the model's enhanced capacity to process fine details at these resolutions.

**Annealing.** In the final stage of pre-training, we perform an annealing phase to sharpen the model's capabilities on a highly curated dataset. This is achieved by training the model for one epoch on a high-quality subset of 20M samples, which were selected from Section B.2 by applying stricter filtering thresholds for aesthetic score, image clarity, semantic similarity, watermark, and so on. This annealing step significantly improves the model's final output, enhancing overall image structure, composition, texture, and aesthetic appeal.

## 3.3 POST-TRAINING

Following pre-training on a broad corpus to establish a generalist model, post-training serves to align the model's output with human preferences and downstream tasks. We achieve this alignment via a two-stage process: Supervised Fine-Tuning (SFT) followed by Direct Preference Optimization (DPO) (Rafailov et al., 2023). The hyperparameters for each stage are in Table 1.

**Supervised Fine-Tuning (SFT).** The SFT stage enhances the model's instruction-following capabilities and aligns its outputs with human preferences. The SFT dataset, comprising a total of 5M

samples, is organized into three components: 1) a corpus of human-selected image-text pairs with high semantic consistency and visual appeal, augmented by images from other generative models to improve the model's handling of complex and imaginative prompts through distillation; 2) Chain-of-Thought (CoT) data (Wei et al., 2022; Deng et al., 2025), improving text-to-image generation by incorporating a language-based reasoning step before the final image is created; 3) high-quality instruction-guided image-to-image data from Section B.3 to strengthen the model's image editing capabilities.

**Direct Preference Optimization (DPO).**    To align our model with human preferences, we employ Direct Preference Optimization (DPO)  (Rafailov et al., 2024), a method inspired by Diffusion-DPO (Wallace et al., 2024).  To this end, we construct two distinct types of preference datasets from a curated set of approximately 20,000 diverse prompts.

1. Standard DPO Dataset: For each prompt $c$, we directly use the SFT model to generate 16 candidate images. These images is then scored by ImageReward (Xu et al., 2023) to form a preference pair $(y^w, y^l)$, where the winning image $y^w$ is randomly sampled from the top 4 candidates, while the losing image $y^l$ is randomly sampled from the remaining 12.

2. Self-CoT DPO Dataset: To enhance the model's reasoning capabilities, we introduce an explicit reasoning step. For each prompt $c$, we first prompt our model to generate a detailed textual CoT, which is then extended to the original prompt. Using this CoT-enhanced prompt, we follow the identical pipeline as above to form a preference pair $(y^w, y^l)$.

### 3.4    DATA

We construct a diverse training corpus designed to foster both robust text-to-image generation and versatile image editing capabilities. Detailed pipelines are described in  Section B.

## 4    MODEL PERFORMANCE

### 4.1    PERFORMANCE OF TEXT-TO-IMAGE GENERATION

We comprehensively evaluate the text-to-image (T2I) generation performance of NextStep-1 on several representative benchmarks, each targeting different aspects of image generation, including visual-textual alignment and world knowledge (full results in section A). As shown in Table 2, we assess NextStep-1's prompt-following ability across three key benchmarks, GenEval (Ghosh et al., 2023) and GenAI-Bench (Li et al., 2024a), and OneIG-Bench (Chang et al., 2025). Results demonstrate that NextStep-1 is competitive with leading diffusion models. Furthermore, its outstanding performance on long-context and multi-object scene, confirms its reliable compositional fidelity under complex prompts.

### 4.2    PERFORMANCE OF IMAGE EDITING

**Quantitative Results on Editing Benchmarks.**    We developed NextStep-1-Edit by finetuning NextStep-1 on 1M high-quality edit-only data in Section B.3, demonstrates competitive performance against advanced diffusion-based models. As shown in Table 3, NextStep-1-Edit achieves scores of **6.58** on GEdit-Bench-EN (Liu et al., 2025c) and **3.71** on ImgEdit-Bench (Ye et al., 2025), indicating its strong practical editing capabilities.

## 5    DISCUSSIONS

### 5.1    WHAT GOVERNS IMAGE GENERATION: THE AR TRANSFORMER OR THE FM HEAD?

A key architectural distinction of our framework is its direct, autoregressive modeling of continuous image tokens using a flow matching objective. Prevailing autoregressive models for image generation  (Sun et al., 2023; 2024b; Dong et al., 2024; Zhou et al., 2025; Chen et al., 2025a) typically rely on heavy, diffusion models for a entire image: an autoregressive model first produces a semantic

Table 2: Comparison of image-text alignment on GenEval (Ghosh et al., 2023), GenAI-Bench (Lin et al., 2024), and DPG-Bench (Hu et al., 2024). * result is with rewriting. † result is with Self-CoT.

| Method | GenEval↑ | GenAI-Bench↑ | | DPG-Bench↑ |
|---|---|---|---|---|
| | | Basic | Advanced | |
| *Proprietary* | | | | |
| DALL-E 3 (Betker et al., 2023) | 0.67 | 0.90 | 0.70 | 83.50 |
| Seedream 3.0 (Gao et al., 2025) | 0.84 | - | - | 88.27 |
| GPT4o (OpenAI, 2025b) | 0.84 | - | - | 85.15 |
| *Diffusion* | | | | |
| Stable Diffusion 1.5 (Rombach et al., 2022) | 0.43 | - | - | - |
| Stable Diffusion XL (Podell et al., 2024) | 0.55 | 0.83 | 0.63 | 74.65 |
| Stable Diffusion 3 Medium (Esser et al., 2024) | 0.74 | 0.88 | 0.65 | 84.08 |
| Stable Diffusion 3.5 Large (Esser et al., 2024) | 0.71 | 0.88 | 0.66 | 83.38 |
| PixArt-Alpha (Chen et al., 2024) | 0.48 | - | - | 71.11 |
| Flux.1-dev (Labs, 2024) | 0.66 | 0.86 | 0.65 | 83.79 |
| Transfusion (Zhou et al., 2025) | 0.63 | - | - | - |
| CogView4 (Z.ai, 2025) | 0.73 | - | - | 85.13 |
| Lumina-Image 2.0 (Qin et al., 2025) | 0.73 | - | - | 87.20 |
| HiDream-I1-Full (Cai et al., 2025) | 0.83 | **0.91** | 0.66 | 85.89 |
| Mogao (Liao et al., 2025) | **0.89** | - | 0.68 | 84.33 |
| BAGEL (Deng et al., 2025) | 0.82/0.88$^\dagger$ | 0.89/0.86$^\dagger$ | 0.69/**0.75**$^\dagger$ | 85.07 |
| Show-o2-7B (Xie et al., 2025b) | 0.76 | - | - | 86.14 |
| OmniGen2 (Wu et al., 2025b) | 0.80/0.86* | - | - | 83.57 |
| Qwen-Image (Wu et al., 2025a) | 0.87 | - | - | **88.32** |
| *AutoRegressive* | | | | |
| SEED-X (Ge et al., 2024) | 0.49 | 0.86 | 0.70 | - |
| Show-o (Xie et al., 2024) | 0.53 | 0.70 | 0.60 | - |
| VILA-U (Wu et al., 2024) | - | 0.76 | 0.64 | - |
| Emu3 (Wang et al., 2024b) | 0.54/0.65* | 0.78 | 0.60 | 80.60 |
| Fluid (Fan et al., 2024) | 0.69 | - | - | - |
| Infinity (Han et al., 2025) | 0.79 | - | - | 86.60 |
| Janus-Pro-7B (Chen et al., 2025b) | 0.80 | 0.86 | 0.66 | 84.19 |
| Token-Shuffle (Ma et al., 2025b) | 0.62 | 0.78 | 0.67 | - |
| NextStep-1 | 0.63/0.73$^\dagger$ | 0.88/0.90$^\dagger$ | 0.67/0.74$^\dagger$ | 85.28 |

Table 3: Comparison of image editing performance on GEdit-Bench (Full Set) (Liu et al., 2025c) and ImgEdit-Bench (Ye et al., 2025). G_SC, G_PQ, and G_O refer to the metrics evaluated by GPT-4.1 (OpenAI, 2025a). Performance is evaluated based on the NextStep-1-Edit with 1:1 aspect ratio.

| Model | GEdit-Bench-EN (Full Set)↑ | | | GEdit-Bench-CN (Full Set)↑ | | | ImgEdit-Bench↑ |
|---|---|---|---|---|---|---|---|
| | G_SC | G_PQ | G_O | G_SC | G_PQ | G_O | |
| *Proprietary* | | | | | | | |
| Gemini 2.0 (Gemini2, 2025) | 6.87 | 7.44 | 6.51 | 5.26 | 7.60 | 5.14 | - |
| Doubao (Shi et al., 2024) | 7.22 | 7.89 | 6.98 | 7.17 | 7.79 | 6.84 | - |
| GPT-4o (OpenAI, 2025b) | **7.74** | **8.13** | **7.49** | 7.52 | 8.02 | 7.30 | **4.20** |
| Flux.1-Kontext-pro (Labs et al., 2025) | 7.02 | 7.60 | 6.56 | 1.11 | 7.36 | 1.23 | - |
| *Open-source* | | | | | | | |
| Instruct-Pix2Pix (Brooks et al., 2023) | 3.30 | 6.19 | 3.22 | - | - | - | 1.88 |
| MagicBrush (Zhang et al., 2023a) | 4.52 | 6.37 | 4.19 | - | - | - | 1.83 |
| AnyEdit (Yu et al., 2024a) | 3.05 | 5.88 | 2.85 | - | - | - | 2.45 |
| OmniGen (Xiao et al., 2024) | 5.88 | 5.87 | 5.01 | - | - | - | 2.96 |
| OmniGen2 (Wu et al., 2025b) | 7.16 | 6.77 | 6.41 | - | - | - | 3.44 |
| Step1X-Edit v1.0 (Liu et al., 2025c) | 7.13 | 7.00 | 6.44 | 7.30 | 7.14 | 6.66 | 3.06 |
| Step1X-Edit v1.1 (Liu et al., 2025c) | **7.66** | **7.35** | **6.97** | 7.65 | 7.40 | 6.98 | - |
| BAGEL (Deng et al., 2025) | 7.36 | 6.83 | 6.52 | 7.34 | 6.85 | 6.50 | 3.42 |
| Flux.1-Kontext-dev (Labs et al., 2025) | - | - | 6.26 | - | - | - | 3.71 |
| GPT-Image-Edit (Wang et al., 2025c) | - | - | 7.24 | - | - | - | **3.80** |
| NextStep-1 | 7.15 | 7.01 | 6.58 | 6.88 | 7.02 | 6.40 | 3.71 |

embedding, which is then used to condition a diffusion model that generates an entire image in a single denoising process. In contrast, our model autoregressively generates the image patch-by-patch, modeling the distribution of each patch with a significantly more lightweight flow matching model. We argue that this establishes our framework under the pure autoregressive paradigm with next-token prediction (NTP) modeling, rather than a diffusion model merely orchestrated by a Transformer.

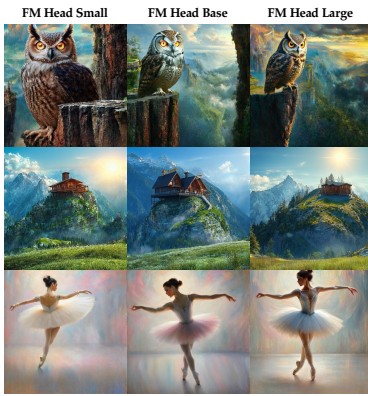

FM Head Small    FM Head Base    FM Head Large

Figure 2: Images generated under different flow-matching heads.

Table 4: Configurations for different flow-matching heads.

|  | Layers | Hidden Size | # Parameters |
|---|---|---|---|
| FM Head Small | 6 | 1024 | 40M |
| FM Head Base | 12 | 1536 | 157M |
| FM Head Large | 24 | 2048 | 528M |

Table 5: Quantitative results for different flow-matching head configurations. All variants are finetuned from the baseline with a newly initialized head.

|  | GenEval | GenAI-Bench | DPG-Bench |
|---|---|---|---|
| Baseline | 0.59 | 0.77 | 85.15 |
| w/ FM Head Small | 0.55 | 0.76 | 83.46 |
| w/ FM Head Base | 0.55 | 0.75 | 84.68 |
| w/ FM Head Large | 0.56 | 0.77 | 85.50 |

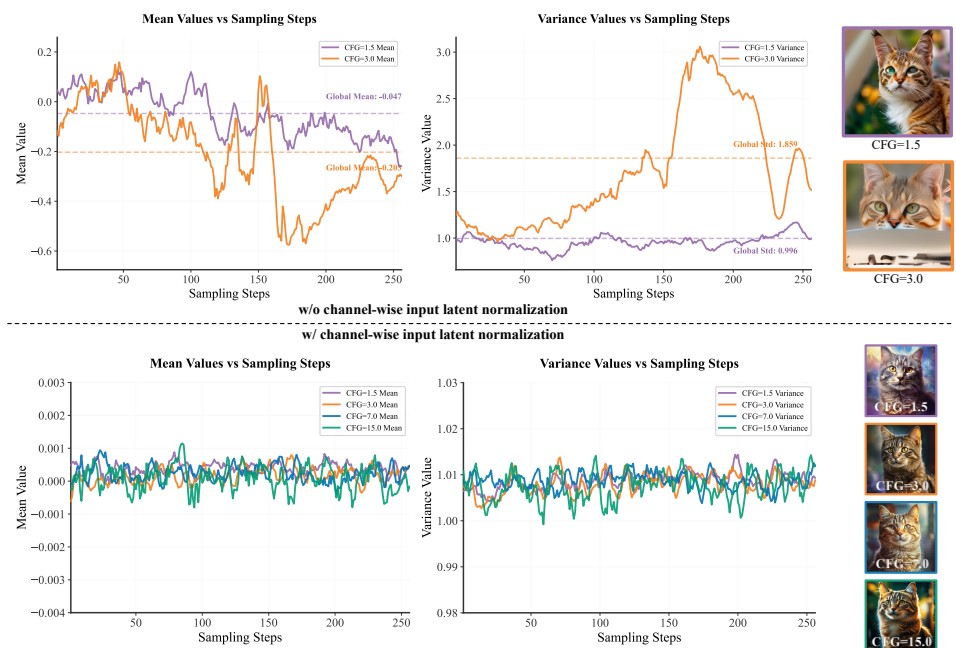

Figure 3: Evolution of per-token mean and variance over sampling steps under two CFG settings. At CFG = 1.5, the mean and variance stay close to 0 and 1, respectively, indicating stability. At CFG = 3.0, they drift significantly, causing image quality degradation. With normalization, the distributions of output latents remain stable across all CFG settings.

A key finding from our experiments is the model's surprising insensitivity to the size of its flow-matching head. We ablated this on three heads with different sizes (small, base, and large). For each experiment, we re-initialized and trained only the head for 10k steps. Despite the significant variation in model size, all three heads produced remarkably similar results (Table 5, Figure 2). This insensitivity to the head's size strongly suggests that the transformer backbone performs the core generative modeling of the conditional distribution $p(x_i \mid x_{<i})$. The flow-matching head, akin to the LM head in language models, primarily acts as a lightweight sampler that translates the transformer's contextual prediction into a continuous token. Consequently, the essential generative logic resides within the transformer's autoregressive NTP process.

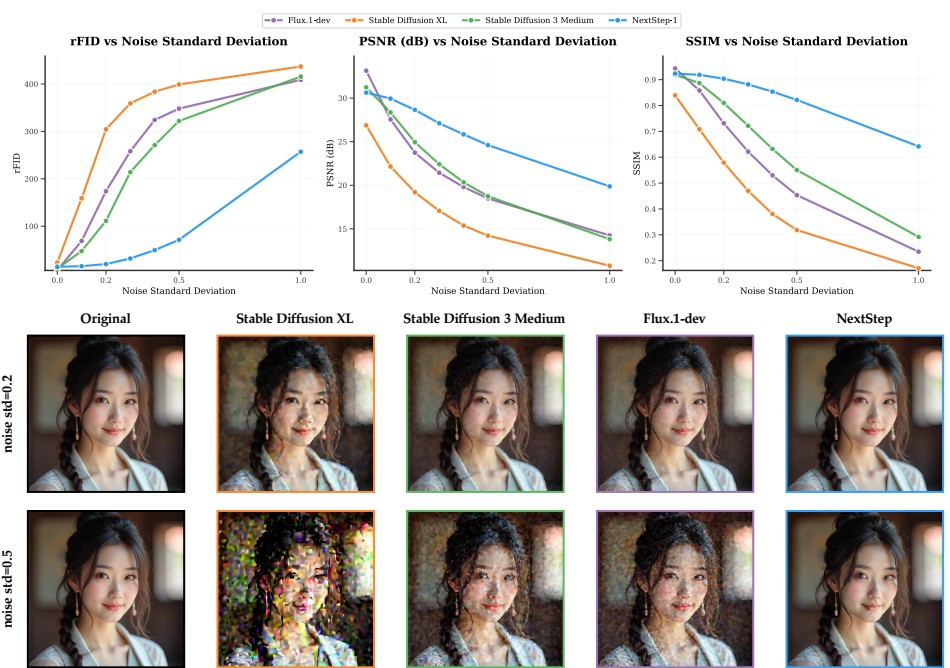

Figure 4: Impact of Noise Perturbation on Image Tokenizer Performance. The top panel displays quantitative metrics (rFID↓, PSNR↑, and SSIM↑) versus noise intensity. The bottom panel presents qualitative reconstruction examples at noise standard deviations of 0.2 and 0.5.

## 5.2 TOKENIZER IS THE KEY TO IMAGE GENERATION

**Mitigating Instability under Strong Classifier-Free Guidance.** A known failure mode in VAE-based autoregressive models is the emergence of visual artifacts, such as gray patches, particularly under strong classifier-free guidance scales (Fan et al., 2024). While prior work hypothesized this instability stemmed from discontinuities in 1D positional embeddings, our analysis reveals that the root cause lies in the amplification of token-level distributional shifts under high guidance scales.

At inference time, CFG is introduced to enhance conditional fidelity. The guided prediction $\tilde{v}$ is computed via an interpolation:

$$\tilde{v}(x|y) = (1 - w) \cdot v_\theta(x|\varnothing) + w \cdot v_\theta(x|y) \tag{4}$$

where $v_\theta(x|\varnothing)$ and $v_\theta(x|y)$ are the unconditional and conditional predictions, and $w$ is guidance scale. In diffusion models, inference with high guidance scale is stable because latent variables are typically normalized, ensuring that conditional and unconditional predictions maintain a **consistent scale**. However, in token-level autoregressive models, global normalization of the entire latent tensor does not enforce per-token statistical consistency. Consequently, small discrepancies between conditional and unconditional predictions are magnified by a large guidance scale, leading to a significant drift in the statistics of generated tokens over the sequence.

We empirically demonstrate this phenomenon in Figure 3. At a moderate guidance scale of 1.5, the per-token mean and variance remain stable throughout the generation process. In contrast, at a high guidance scale of 3.0, both statistics diverge significantly for later tokens, a distributional shift that corresponds directly to the appearance of visual artifacts. Our tokenizer design, which incorporates token-wise normalization (see Equation (3)), directly addresses this issue by enforcing per-token statistical stability. This simple but critical design choice mitigates the instability, enabling the use of strong guidance without degrading image quality.

**A Regularized Latent Space is Critical for Generation** A key finding of our work is a counter-intuitive inverse correlation between the generation loss and the final synthesis quality of the autoregressive model. Specifically, applying higher noise intensity ($\gamma$ in Equation (3)) during tokenizer training increases generation loss but paradoxically improves the quality of the generated images.

Table 6: Comparison of reconstruction performance on ImageNet-1K 256×256 (Deng et al., 2009).

| Tokenizer | Latent Shape | PSNR ↑ | SSIM ↑ |
|---|---|---|---|
| *Discrete Tokenizer* | | | |
| SBER-MoVQGAN (270M) (Zheng et al., 2022) | 32x32 | 27.04 | 0.74 |
| LlamaGen (Sun et al., 2024a) | 32x32 | 24.44 | 0.77 |
| VAR (Tian et al., 2024) | 680 | 22.12 | 0.62 |
| TiTok-S-128 (Yu et al., 2024b) | 128 | 17.52 | 0.44 |
| Sefltok (Wang et al., 2025b) | 1024 | 26.30 | 0.81 |
| *Continuous Tokenizer* | | | |
| Stable Diffusion 1.5 (Rombach et al., 2022) | 32x32x4 | 25.18 | 0.73 |
| Stable Diffusion XL (Podell et al., 2024) | 32x32x4 | 26.22 | 0.77 |
| Stable Diffusion 3 Medium (Esser et al., 2024) | 32x32x16 | 30.00 | 0.88 |
| Flux.1-dev (Labs, 2024) | 32x32x16 | **31.64** | **0.91** |
| **NextStep-1** | 32x32x16 | 30.60 | 0.89 |

For instance, NextStep-1 uses a tokenizer trained at $\gamma = 0.5$, which incurred the highest generation loss yet produced the highest-fidelity images. Conversely, tokenizers trained for low generation loss caused the autoregressive model to yield outputs resembling pure noise.

We attribute this phenomenon to noise regularization cultivating a well-conditioned latent space. This process enhances two key properties: the tokenizer decoder's robustness to latent perturbations (Figure 4) and a more dispersed latent distribution (Figure A3), a property prior work has also found beneficial for generation (Yang et al., 2025; Yao et al., 2025; Sun et al., 2024c). While it remains unclear whether robustness or dispersion plays a critical role, these results underscore the practical benefits of noise-based regularization and highlight promising directions for future analysis.

**Reconstruction Quality is the Upper Bound of Generation Quality.** The reconstruction fidelity of the image tokenizer fundamentally determines the upper bound for the quality of the final generated image, particularly for fine details and textures. This principle has been validated in numerous recent studies (Esser et al., 2024; Labs, 2024; Dai et al., 2023), leading to a trend in the diffusion paradigm of building generative models on top of VAEs with exceptional reconstruction performance (e.g., PSNR >30). In contrast, VQ-based autoregressive models have historically struggled to surpass this threshold, as shown in Table 6. While a trade-off between reconstruction and generation quality is often debated (Yao et al., 2025), our work successfully applies autoregressive models to high-fidelity continuous VAEs, bridging this gap.

## 6 RELATED WORK

**Diffusion and Flow Matching Models.** Diffusion models have established themselves as the dominant paradigm for high-fidelity image synthesis. While early foundations relied on U-Net (Ronneberger et al., 2015) architectures, recent advancements have shifted towards Diffusion Transformers (DiTs) (Peebles and Xie, 2023) to leverage the scalability of attention mechanisms. More recently, Flow Matching (Lipman et al., 2023a; Liu et al., 2022) has been adopted to rectify generation trajectories, with models like Flux (Labs, 2024) demonstrating SOTA visual quality. However, these systems typically rely on separate, pre-trained text encoders (e.g., T5-XXL(Raffel et al., 2020), CLIP(Radford et al., 2021)) to process prompts. This decoupled design imposes a fixed context window, limiting deep multimodal fusion and the model's ability to handle interleaved inputs or long-context reasoning naturally. Furthermore, the generation process in these models—often involving bidirectional attention over the entire image latent space—is computationally intensive and distinct from the token-by-token reasoning inherent to Large Language Models (LLMs).

**Discrete Autoregressive Models.** Another line of research treats image generation as a sequence modeling problem, aiming to unify vision and language under a single transformer backbone. Pioneering works such as LlamaGen (Sun et al., 2024a), Janus-Pro (Chen et al., 2025b), Emu3 (Wang et al., 2024b), and InfinityStar (Liu et al., 2025a) have successfully adapted the "next-token prediction" paradigm to vision. However, these methods predominantly rely on Vector Quantization (VQ) (Zheng et al., 2022) to discretize continuous images into a finite codebook. This introduces a fun-

damental bottleneck: the discretization process incurs information loss, limiting the reconstruction upper bound.

**Unified and Hybrid Architectures.** To bridge the gap between the generative quality of diffusion and the scalability of LLMs, recent studies have explored hybrid architectures. Models like Dream-LLM (Dong et al., 2024), Emu2 (Sun et al., 2024b), and Qwen-Image(Wu et al., 2025a) achieve SOTA generation by appending a heavy diffusion head (often a MMDiT (Esser et al., 2024)) to an LLM. While effective, relying on the separate text encoder and diffusion model limits the potential for unified multi-modality fusion and making performance sensitive to the interleaved generation. Another direction, represented by Transfusion (Zhou et al., 2025) and Bagel (Deng et al., 2025), integrates continuous noisy inputs and diffusion loss directly into the LLM training. However, these methods often compromise the "pure" autoregressive nature of LLMs for image data. They typically employ bidirectional attention and require feeding both the noisy latent and the clean image (or condition) into the model simultaneously. These special designs increase the attention cost and sequence length during training, thereby hindering the efficient application of scaling laws compared to pure next-token prediction.

In contrast to the aforementioned approaches, NextStep-1 adheres to a strict causal autoregressive objective on continuous image tokens (Li et al., 2024c; Fan et al., 2025). NextStep-1 demonstrates that a standard LLM backbone, equipped with a lightweight flow-matching head, can serve as the primary generative engine. This approach avoids the quantization artifacts of discrete AR models while maintaining the causal reasoning benefits and infrastructure compatibility of LLMs. By eliminating the need for bidirectional attention or noisy input duplication, NextStep-1 achieves high-fidelity generation with a unified, scalable, and simplified architecture.

## 7 CONCLUSION

We propose NextStep-1, a fully autoregressive model with versatile image generation and editing capabilities that sequentially predicts the next continuous image patch token via a lightweight flow matching head. To address the stability and quality issues associated with high-dimensional continuous image tokens, we develop a robust autoencoder using noise perturbation and token-wise input latent normalization. We show that this design is crucial for the performance of autoregressive model, and demonstrate its competitive results across a wide variety of image generation and editing tasks, not only achieving state-of-the-art results among existing autoregressive image generation models, but also showing competitive performance when compared to leading diffusion-based methods. We plan to release the model and code to inspire further research, foster collaboration, and accelerate progress in this exciting frontier.

### ACKNOWLEDGMENTS

We thank Ailin Huang, Bin Wang, Changxin Miao, Deshan Sun, En Yu, Fukun Yin, Gang Yu, Hao Nie, Haoran Lv, Hanpeng Hu, Jia Wang, Jian Zhou, Jianjian Sun, Kaijun Tan, Kang An, Kangheng Lin, Liang Zhao, Mei Chen, Peng Xing, Rui Wang, Shiyu Liu, Shutao Xia, Tianhao You, Wei Ji, Xianfang Zeng, Xin Han, Xuelin Zhang, Yana Wei, Yanming Xu, Yimin Jiang, Yingming Wang, Yu Zhou, Yucheng Han and Ziyang Meng from StepFun for their strong support.

We would also like to sincerely thank Tianhong Li and Yonglong Tian for their insightful discussions.

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

APPENDIX

# A    MORE RESULTS

**Qualitative Results.** Figure A1 visually demonstrates NextStep-1's ability to generate diverse, high-quality images with fine details, perform precise edits, and carry out complex free-form manipulations that alter context and actions while preserving subject consistency.

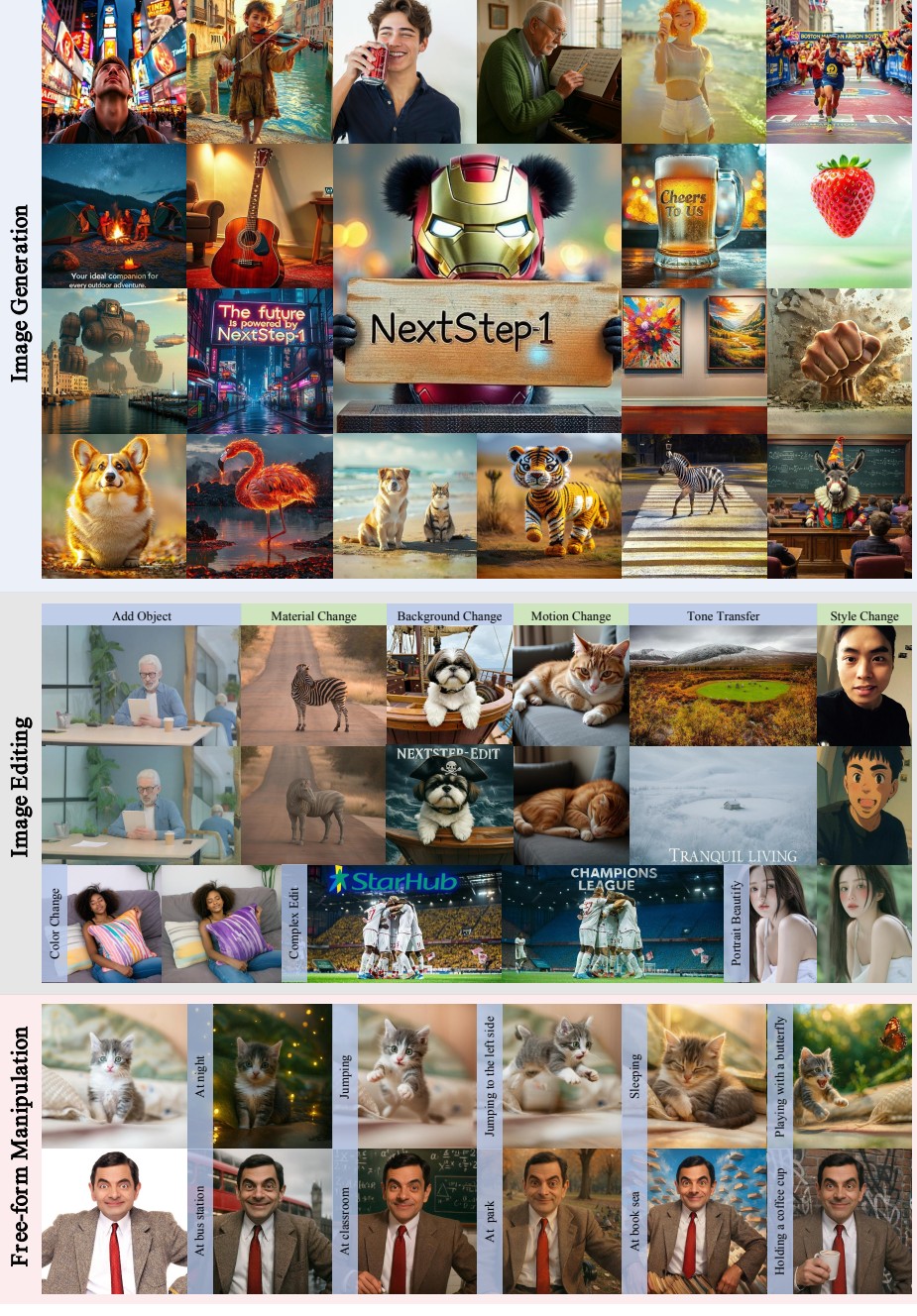

Figure A1: Overview of NextStep-1 in high-fidelity image generation, diverse image editing, and complex free-form manipulation.

Table A1: Comparison on OneIG-Bench (Chang et al., 2025) in English prompts.

| Method | Alignment | Text | Reasoning | Style | Diversity | Overall↑ |
|---|---|---|---|---|---|---|
| *Proprietary* | | | | | | |
| Imagen3 (Baldridge et al., 2024) | 0.843 | 0.343 | 0.313 | 0.359 | 0.188 | 0.409 |
| Recraft V3 (team, 2024) | 0.810 | 0.795 | 0.323 | 0.378 | 0.205 | 0.502 |
| Kolors 2.0 (team, 2025) | 0.820 | 0.427 | 0.262 | 0.360 | 0.300 | 0.434 |
| Seedream 3.0 (Gao et al., 2025) | 0.818 | 0.865 | 0.275 | 0.413 | 0.277 | 0.530 |
| Imagen4 (deepmind Imagen4 team, 2025) | 0.857 | 0.805 | 0.338 | 0.377 | 0.199 | 0.515 |
| GPT-4o (OpenAI, 2025b) | 0.851 | 0.857 | **0.345** | **0.462** | 0.151 | 0.533 |
| *Diffusion* | | | | | | |
| Stable Diffusion 1.5 (Rombach et al., 2022) | 0.565 | 0.010 | 0.207 | 0.383 | **0.429** | 0.319 |
| Stable Diffusion XL (Podell et al., 2024) | 0.688 | 0.029 | 0.237 | 0.332 | 0.296 | 0.316 |
| Stable Diffusion 3.5 Large (Stability-AI, 2024) | 0.809 | 0.629 | 0.294 | 0.353 | 0.225 | 0.462 |
| Flux.1-dev (Labs, 2024) | 0.786 | 0.523 | 0.253 | 0.368 | 0.238 | 0.434 |
| CogView4 (Z.ai, 2025) | 0.786 | 0.641 | 0.246 | 0.353 | 0.205 | 0.446 |
| SANA-1.5 1.6B (PAG) (Xie et al., 2025a) | 0.762 | 0.054 | 0.209 | 0.387 | 0.222 | 0.327 |
| SANA-1.5 4.8B (PAG) (Xie et al., 2025a) | 0.765 | 0.069 | 0.217 | 0.401 | 0.216 | 0.334 |
| Lumina-Image 2.0 (Qin et al., 2025) | 0.819 | 0.106 | 0.270 | 0.354 | 0.216 | 0.353 |
| HiDream-I1-Full (Cai et al., 2025) | 0.829 | 0.707 | 0.317 | 0.347 | 0.186 | 0.477 |
| BLIP3-o (Chen et al., 2025a) | 0.711 | 0.013 | 0.223 | 0.361 | 0.229 | 0.307 |
| BAGEL (Deng et al., 2025) | 0.769 | 0.244 | 0.173 | 0.367 | 0.251 | 0.361 |
| Show-o2-1.5B (Xie et al., 2025b) | 0.798 | 0.002 | 0.219 | 0.317 | 0.186 | 0.304 |
| Show-o2-7B (Xie et al., 2025b) | 0.817 | 0.002 | 0.226 | 0.317 | 0.177 | 0.308 |
| OmniGen2 (Wu et al., 2025b) | 0.804 | 0.680 | 0.271 | 0.377 | 0.242 | 0.475 |
| Qwen-Image (Wu et al., 2025a) | **0.882** | **0.891** | 0.306 | 0.418 | 0.197 | **0.539** |
| *AutoRegressive* | | | | | | |
| Emu3 (Wang et al., 2024b) | 0.737 | 0.010 | 0.193 | 0.361 | 0.251 | 0.311 |
| Janus-Pro (Chen et al., 2025b) | 0.553 | 0.001 | 0.139 | 0.276 | 0.365 | 0.267 |
| NextStep-1 | 0.826 | 0.507 | 0.224 | 0.332 | 0.199 | 0.417 |

Table A2: Comparison of world knowledge reasoning on WISE (Niu et al., 2025). † result is with Self-CoT.

| Model | Cultural | Time | Space | Biology | Physics | Chemistry | Overall↑ | Overall (Rewrite)↑ |
|---|---|---|---|---|---|---|---|---|
| *Proprietary* | | | | | | | | |
| GPT-4o (OpenAI, 2025b) | **0.81** | **0.71** | **0.89** | **0.83** | **0.79** | **0.74** | **0.80** | - |
| *Diffusion* | | | | | | | | |
| Stable Diffusion 1.5 (Rombach et al., 2022) | 0.34 | 0.35 | 0.32 | 0.28 | 0.29 | 0.21 | 0.32 | 0.50 |
| Stable Diffusion XL (Podell et al., 2024) | 0.43 | 0.48 | 0.47 | 0.44 | 0.45 | 0.27 | 0.43 | 0.65 |
| Stable Diffusion 3.5 Large (Stability-AI, 2024) | 0.44 | 0.50 | 0.58 | 0.44 | 0.52 | 0.31 | 0.46 | 0.72 |
| PixArt-Alpha (Chen et al., 2024) | 0.45 | 0.50 | 0.48 | 0.49 | 0.56 | 0.34 | 0.47 | 0.63 |
| Playground v2.5 (Li et al., 2024b) | 0.49 | 0.58 | 0.55 | 0.43 | 0.48 | 0.33 | 0.49 | 0.71 |
| Flux.1-dev (Labs, 2024) | 0.48 | 0.58 | 0.62 | 0.42 | 0.51 | 0.35 | 0.50 | 0.73 |
| MetaQuery-XL (Pan et al., 2025) | 0.56 | 0.55 | 0.62 | 0.49 | 0.63 | 0.41 | 0.55 | - |
| BAGEL (Deng et al., 2025) | 0.44/0.76† | 0.55/0.69† | 0.68/0.75† | 0.44/0.65† | 0.60/0.75† | 0.39/0.58† | 0.52/0.70† | 0.71/0.77† |
| Qwen-Image (Wu et al., 2025a) | 0.62 | 0.63 | 0.77 | 0.57 | 0.75 | 0.40 | 0.62 | - |
| *AutoRegressive* | | | | | | | | |
| Show-o-512 (Xie et al., 2024) | 0.28 | 0.40 | 0.48 | 0.30 | 0.46 | 0.30 | 0.35 | 0.64 |
| VILA-U (Wu et al., 2024) | 0.26 | 0.33 | 0.37 | 0.35 | 0.39 | 0.23 | 0.31 | - |
| Emu3 (Wang et al., 2024b) | 0.34 | 0.45 | 0.48 | 0.41 | 0.45 | 0.27 | 0.39 | 0.63 |
| Janus-Pro-7B (Chen et al., 2025b) | 0.30 | 0.37 | 0.49 | 0.36 | 0.42 | 0.26 | 0.35 | 0.71 |
| **NextStep-1** | 0.51/0.70 | 0.54/0.65 | 0.61/0.69 | 0.52/0.63 | 0.63/0.73 | 0.48/0.52† | 0.54/0.67 | **0.79/0.83**† |

**Quantitative Results.** We also conduct a fine-grained analysis on OneIG-Bench (Chang et al., 2025) with English prompts. Table A1 reflects NextStep-1's significant advantage across areas such as alignment, text rendering, reasoning and stylistic control over existing autoregressive models. To evaluate NextStep-1's ability to integrate world knowledge into image generation, we use the WISE benchmark (Niu et al., 2025), which emphasizes factual grounding and semantic understanding. As shown in Table A2, NextStep-1 achieves the best performance among autoregressive models, also exceeding most diffusion models, with an even more significant boost under the prompt rewrite protocol. Collectively, these results demonstrate NextStep-1's robust knowledge-aware semantic alignment and cross-domain reasoning capabilities.

## B   DATA PIPELINES

To comprehensively equip our model with broad and versatile capabilities, we construct a diverse training corpus composed of four primary data categories: a text-only corpus, image-text pair data, image-to-image data, and interleaved data. Each category is curated to serve a distinct role in fostering different aspects of the model's generative abilities.

### B.1   TEXT-ONLY CORPUS

To preserve the extensive language capabilities inherent in the large language model (LLM), we incorporate **400B text-only tokens** sampled from Step-3 (Wang et al., 2025a) during training.

### B.2   IMAGE-TEXT PAIR DATA

Data consisting of image-text pairs forms the foundation of the model's text-to-image generation capabilities. We developed a comprehensive pipeline to curate a high-quality, large-scale dataset from a diverse set of initial sources.

1. Data Sourcing: We collected a large-scale dataset from diverse sources, including web data, multi-task VQA data and text-rich documents.

2. Quality-Based Filtering: We then applied a rigorous filtering process, evaluating each image on aesthetic quality, watermark presence, clarity, OCR detection, and text-image semantic alignment.

3. Re-captioning: After deduplicating the filtered images, we used the Step-1o-turbo to generate rich and detailed captions for each image in both English and Chinese.

This multi-stage pipeline yields a final dataset of **550M high-quality image-text pairs**, providing a foundation for training a model with both strong aesthetic sense and broad world knowledge.

### B.3   INSTRUCTION-GUIDED IMAGE-TO-IMAGE DATA

To enable a wide range of practical applications, we curated a high-quality dataset for instruction-guided image-to-image tasks, such as visual perception (Kirillov et al., 2023), controllable image generation (Zhang et al., 2023b), image restoration (Labs, 2025), general image editing (Peng et al., 2024), and more.

For visual perception and controllable image generation tasks, we synthesized 1M samples by applying the annotator of ControlNet (Zhang et al., 2023b) to a part of our high-quality image-text pair data. For image restoration and general image editing, we collected 3.5M samples, comprising data from GPT-Image-Edit (Wang et al., 2025c), Step1X-Edit (Liu et al., 2025c), and a proprietary in-house dataset. Following Step1X-Edit (Liu et al., 2025c), all editing data were subjected to a rigorous VLM-based filtering pipeline that assessed both image-pair quality, rationality, consistency, and instruction alignment, resulting in about **1M high-quality instruction-guided image-to-image data** for training.

### B.4   INTERLEAVED DATA

Interleaved data seamlessly integrates text and images, offering rich and nuanced sequential associations between modalities. Specifically, our knowledge-rich interleaved dataset is primarily composed of four distinct categories: general video-interleaved data, tutorials, character-centric scenes, and multi-view data.

To endow our model with extensive world knowledge, we first constructed a large-scale, 80M-sample video-interleaved dataset. This was achieved through a meticulous curation pipeline, inspired by Step-Video (Ma et al., 2025a), which encompasses frame extraction, deduplication, and captioning. Furthermore, following the methodology of mmtextbook (Zhang et al., 2025), we collected and processed tutorial videos by leveraging ASR and OCR tools. This component specifically targets text-rich real-world scenes, enhancing the model's textual understanding and generation in

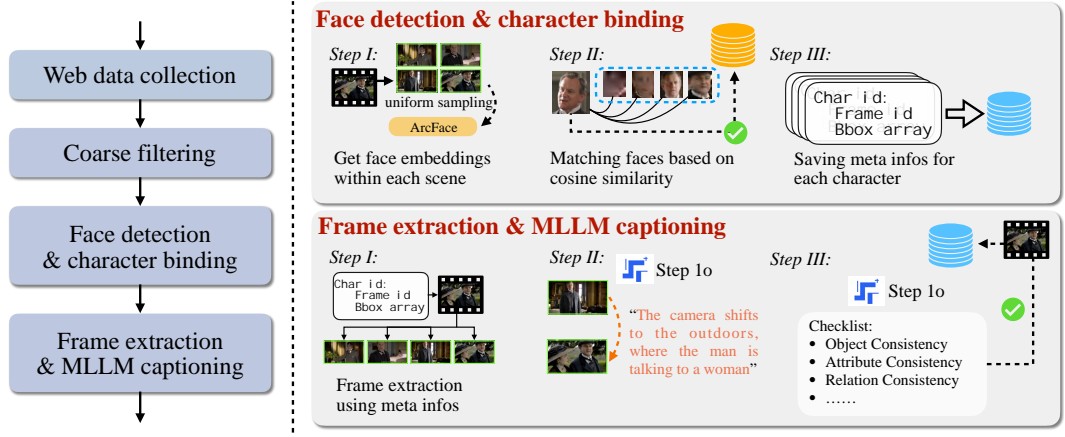

Figure A2: Data processing of character-centric data.

context. A key contribution, detailed in Figure A2, is our character-centric dataset, **NextStep-Video-Interleave-5M**. For this dataset, we extracted video frames centered around specific characters and generated rich, storytelling-style captions akin to (Oliveira and de Matos, 2025), thereby significantly improving the model's capacity for multi-turn interaction. Finally, to bolster geometric reasoning, we curated multiview data from two open-source datasets, MV-ImageNet-v2 (Han et al., 2024) and Objaverse-XL (Deitke et al., 2023), which enhances the model's ability to maintain multiview consistency.

## C  IMAGE TOKENIZER

Our image tokenizer is initialized from the Flux.1-dev VAE (Labs, 2024), selected for its strong reconstruction performance. Differently, as shown in Figure 2, we apply a token-wise normalization operation on the Encoder output, and adopt the noise schedule like $\sigma$-VAE Sun et al. (2024c) during tuning. For better understanding, the normalization operation is detailed in the code of Algorithm 1..

We collect latent distributions in 16 channels for three VAE variants in Figure A3. NextStep-1 VAE aligns best with the normal distribution, reflecting a dispersed latent distribution.

Algorithm 1: Python code for the Token-wise Normalization of NextStep-VAE.

```python
# X: input latent tensor of shape (B, H, W, C)
# eps: stability constant, default 1e-6
def normalization_operation(X, eps=1e-6):
    # Compute mean and std along the channel dimension (-1)
    mu = X.mean(dim=-1, keepdim=True)
    sigma = X.std(dim=-1, keepdim=True, unbiased=False)

    # Apply normalization: zero-mean and unit-variance
    X_norm = (X - mu) / (sigma + eps)

    return X_norm
```

## D  LIMITATIONS AND CHALLENGES

**Artifacts.**    While NextStep-1 successfully demonstrates that autoregressive models can operate on high-dimensional continuous latent spaces, achieving generation quality comparable to diffusion models, this approach also introduces unique stability challenges. We observed the emergence of several distinct generative artifacts when transitioning from a VAE with a lower-dimensional latent space (e.g., spatial downsample factor is 8 and number of latents channel is 4) to one with a higher-

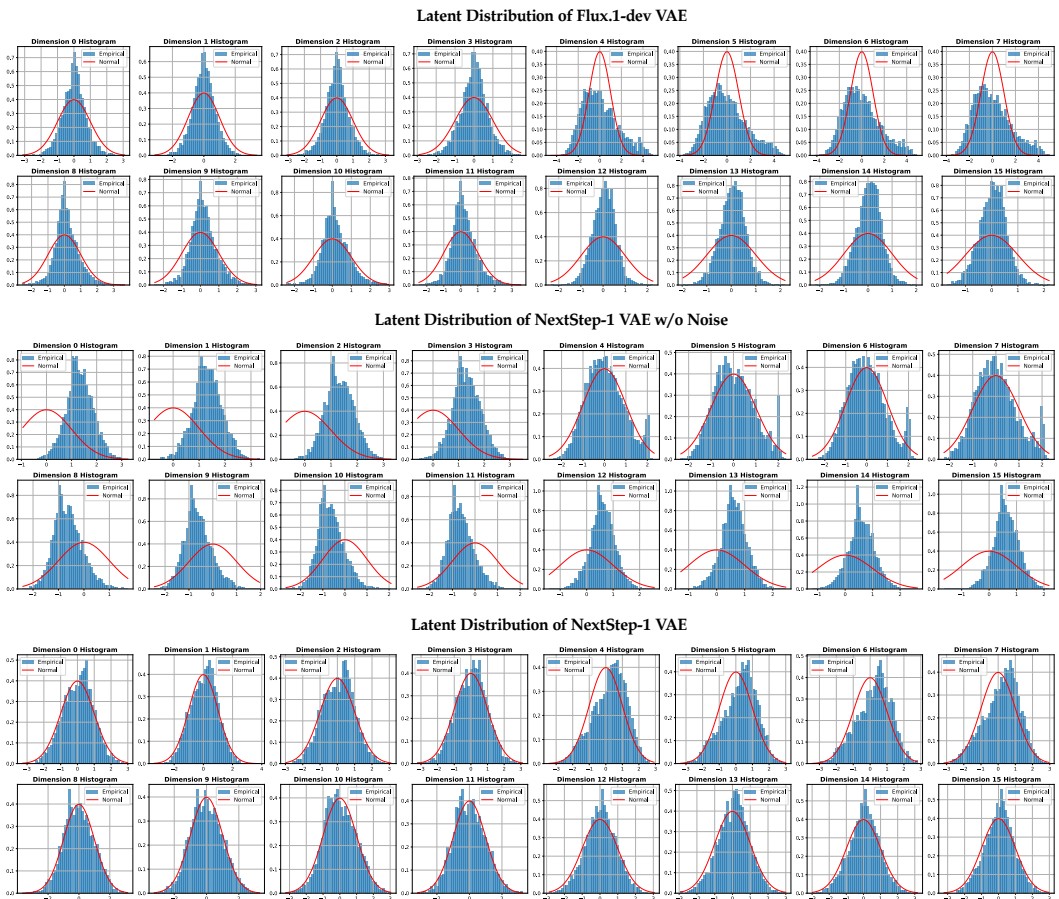

Figure A3: Latent distributions in 16 channels for three VAE variants: Flux.1-dev, NextStep-1 w/o noise, and NextStep-1. Blue bars show empirical histograms; red lines indicate the standard normal distribution.

dimensional space (e.g., spatial downsample factor is 8 and number of latents channel is 16). While the former configuration produced stable outputs, the latter occasionally exhibited failure modes, as illustrated in Figure A4.

While the underlying causes remain an open question, we identify several plausible contributing factors: (1) **Local noise or block-shaped artifacts** emerging in the later stages of generation may arise from numerical instabilities; (2) **Global noise across the image** may reflect under-convergence, implying that additional training could mitigate the issue; and (3) **Subtle grid-like artifacts** could reveal limitations of the 1D positional encoding in capturing 2D spatial relationships.

**Mitigating Artifacts in NextStep 1.1.** Bagel (Deng et al., 2025) is trained on approximately 5T tokens, significantly exceeding the 2T tokens used for NextStep-1. Motivated by this, we incorporated additional high-quality data to continue training NextStep-1, yielding NextStep-1.1. As shown in Figure A5, by leveraging superior data over extended training iterations (approximately 500K) and an improved version of Flow-GRPO (Liu et al., 2025b) in NextStep-1.1, most artifacts observed in NextStep-1 are successfully eliminated.

**Inference Latency of Sequential Decoding.** A theoretical analysis of per-token latency on an H100 GPU (983 TFLOPS, 3.36 TB/s bandwidth) with a batch size of 1, as detailed in Table A3, decomposes the contributions of individual components. The results show that the dominant bottleneck lies in the serial decoding of the LLM, while the multi-step sampling in the flow-matching head also constitutes a substantial portion of the per-token generation cost. These observations suggest two promising directions for accelerating inference. First, the efficiency of the flow matching head

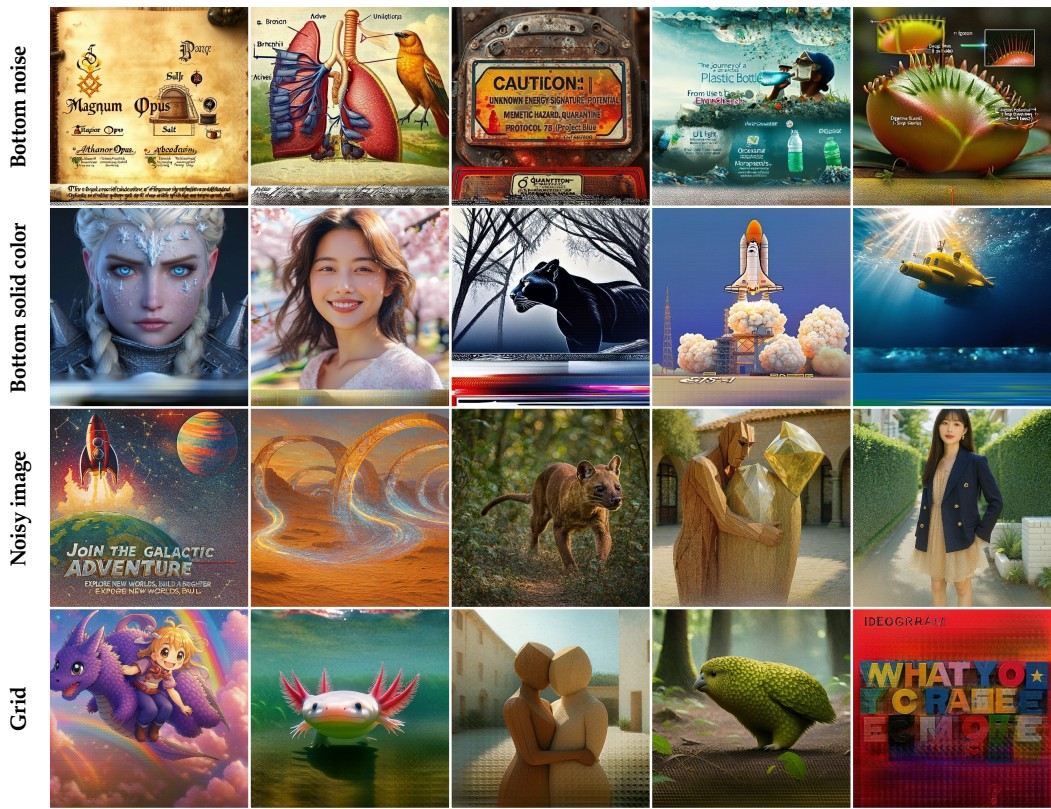

Figure A4: Failure cases for high-dimensional continuous tokens.

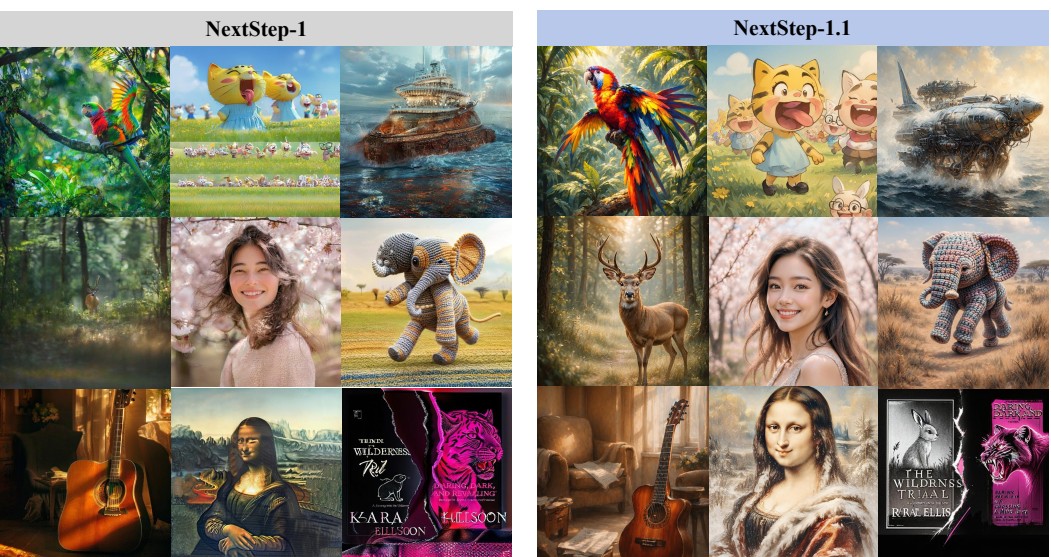

Figure A5: Comparisons between NextStep-1 and NextStep-1.1.

could be improved by reducing its parameter count, applying distillation to achieve few-step generation (Meng et al., 2023), or using more advanced few-step samplers (Lu et al., 2022; 2025). Second, the autoregressive backbone could be accelerated by adapting recent advances from the LLM field, such as speculative decoding (Leviathan et al., 2023) or multi-token prediction (Gloeckle et al., 2024), to the domain of image token generation.

Table A3: Inference latency breakdown at 983 TFLOP/s compute and 3.36 TB/s memory bandwidth.

| Sequence Length | Last-token Latency (ms) | | | Accumulated Latency (s) | |
|---|---|---|---|---|---|
| | LLM Decoder | LM Head | FM Head | Total | w/o FM Head |
| 256 | 7.20 | 0.40 | 3.40 | 2.82 | 1.95 |
| 1024 | 7.23 | 0.40 | 3.40 | 11.31 | 7.83 |
| 4096 | 7.39 | 0.40 | 3.40 | 45.77 | 31.86 |

**Challenges in High-Resolution Training.** Our framework faces two primary challenges in scaling to high-resolution image generation, particularly when compared to diffusion models, which benefit from well-established techniques (Esser et al., 2024; Chen et al., 2024) in this domain. First, the strictly sequential nature of autoregressive generation requires substantially more training steps to converge at higher resolutions. In contrast, diffusion models refine the entire image in parallel at each iteration, enabling more direct exploitation of 2D spatial inductive biases. Second, techniques recently developed for high-resolution diffusion models, such as timestep shift, are difficult to adapt to our setting. This limitation arises because the Flow Matching Head acts primarily as a lightweight sampler, while the transformer backbone performs the core generative modeling; thus, modifications to the sampling process have only marginal impact on the final output. Designing high-resolution generation strategies specifically for patch-wise autoregressive models remains an important direction for future research.

**Challenges in SFT.** SFT in our autoregressive framework poses unique challenges compared to diffusion models. **We observe that fine-tuning on small, high-quality datasets exhibits unstable dynamics.** In contrast to diffusion models, which can often adapt to a target distribution and maintain stable and general image generation with only a few thousand samples, our SFT process yields substantial improvements only when trained on datasets at the million-sample scale. With smaller datasets, the model remains in a precarious equilibrium; it either improves marginally with negligible impact or abruptly overfits to the target distribution. Consequently, identifying an intermediate checkpoint that achieves alignment with the target distribution while preserving general generative capability remains a significant challenge.

# E  LLM USAGE

This paper only uses large language models (LLMs) for language polishing and improving the clarity of writing. No part of the research design, experiments, data analysis, or results was generated by LLMs.

