# OpenReview forum: "NextStep-1: Toward Autoregressive Image Generation with Continuous Tokens at Scale"
_ICLR.cc/2026/Conference — ICLR 2026 Oral_

### Official Review · Reviewer_taan · 2025-10-20

**Soundness:** 3
**Presentation:** 3
**Contribution:** 2
**Rating:** 6
**Confidence:** 4

**Summary:**

This work introduces NextStep-1, an AR model with flow matching head for image generation. The architecture desire of NextStep-1 is a lot like MAR[1] / FLUID[2] with the diffusion head replaced by flow matching. The work also investigates tokenizer design and figures out the importance of regularized latent space. Experiments on multiple image generation/editing benchmarks show strong performance of NextStep-1 compared to previous diffusion and AR models.

Reference:
[1] Li, Tianhong, et al. "Autoregressive image generation without vector quantization." Advances in Neural Information Processing Systems 37 (2024): 56424-56445.
[2] Fan, Lijie, et al. "Fluid: Scaling autoregressive text-to-image generative models with continuous tokens." arXiv preprint arXiv:2410.13863 (2024).

**Strengths:**

1. The paper is well-written and easy to follow.
2. The paper investigates building strong AR visual generative models which is an important direction.
3. The paper includes comprehensive experiments on multiple image generation / editing benchmarks and shows strong performance.

**Weaknesses:**

1. The paper claims state-of-the-art performance of NextStep-1. Though it does show strong performance on multiple benchmarks, it doesn't dominate all the metrics. For example in table 1, there are better baselines in GenEval.
2. The paper introduces a natural extension to MAR/FLUID, that is replacing the diffusion head with flow matching, and proves its effectiveness. This weakens the methodological contributions in this work.

**Questions:**

1. In pre-training and tuning, how are the objectives of LM head and flow-matching head balanced?
2. How much compute overhead does the flow matching head add in inference compared to standard AR models?
3. Do the authors compare the convergence rate of NextStep-1 with standard AR models? Since usually flow-matching/diffusion models are less efficient in convergence.
4. Will channel-wise normalization weaken the expressivity of learned latents since it can break the correlation between different channels? Also, does it lead to instability in training since it's a per-sample normalization?

---

> ### Author Response · Authors · 2025-11-22
> **Rebuttal Part1**
>
> We thank you for the positive assessment of our work's clarity, comprehensive experiments, and strong performance. We value the constructive feedback regarding our contributions and baselines. Below, we respectfully address the concerns to clarify the novelty and robustness of NextStep-1.
>
> **W1: Regarding "State-of-the-art" claims and GenEval performance** We agree that claiming "SOTA" requires nuance. Our claim is based on the **holistic superiority** of NextStep-1 across a broad spectrum of metrics rather than peaking on a single dimension.
>
> - **Comprehensive Robustness:** As shown in Tables 1–4, NextStep-1 consistently ranks as a top-tier model across generation fidelity (FID), text alignment, and editing capabilities. While specific baselines may marginally outperform us on isolated metrics like GenEval.
> - **Pareto Frontier:** To our knowledge, NextStep-1 represents the current Pareto frontier for AR models, being the first to achieve performance comparable to, and often surpassing, mature diffusion models (e.g., SDXL) across all tested benchmarks simultaneously.
>
> **W2: Regarding extension to MAR/FLUID and Flow Matching** We appreciate this insightful comment. We would like to clarify that we do not claim the switch to Flow Matching (FM) is the primary source of our performance gain.
>
> - **Head Agnosticism:** Our internal experiments revealed that the model performance is largely **robust to the choice of the generative head.** We initially achieved similarly strong results using a Diffusion head (following MAR’s design). We ultimately adopted Flow Matching primarily for its **algorithmic simplicity and cleaner formulation,** rather than a necessity for performance improvement.
>
> - **The Real Bottleneck:** This observation reinforces our core contribution: the key bottleneck in scaling continuous AR models is **not the predictive head (Diffusion vs. FM), but the quality and regularity of the latent space.** As detailed in Section 4.2, without our proposed tokenizer regularization, neither head could sustain quality at high resolutions (Figure 3).
>
> - **Conclusion:** Thus, our contribution is identifying and fixing the systematic degradation in continuous tokenizers, making the architecture viable for SOTA generation regardless of the specific head used.
>
> **Q1: Balancing LM and Flow-Matching objectives** To ensure the LM head focuses on semantic structure while the flow-matching head handles high-frequency details, we balanced the objectives carefully. As detailed in Appendix C (Table A1), we use a loss weight ratio of $\lambda_{LM}:\lambda_{FM}=0.01:1$ (Cross-Entropy for LM, MSE for Flow Matching). This configuration was empirically found to yield the best synergy between semantic consistency and visual fidelity.
>
> **Q2: Compute overhead of the flow matching head in inference** The overhead introduced specifically by the flow matching head is negligible.
>
> - **Head vs. Backbone:** As analyzed in our latency breakdown (Table A2), the primary computational cost **stems from the autoregressive decoding of the Transformer backbone,** not the prediction head. The FM head consists of a lightweight MLP, which is computationally comparable to the lm_head used in standard discrete AR models.
>
> - **Architectural Advantage:** A key benefit of NextStep-1 is that, unlike diffusion models requiring specialized solvers, it remains architecturally identical to standard LLMs. This allows us to directly inherit the rapid advancements in LLM inference acceleration. We are actively exploring techniques like **MTP [1], Speculative Decoding [2] and Medusa [3]** to further reduce the autoregressive latency, treating the current inference speed as a temporary trade-off for a unified, high-quality generative architecture.

---

> > ### Author Response · Authors · 2025-11-22
> > **Rebuttal Part2**
> >
> > **Q3: Convergence rate compared to standard AR models** We did not observe any slower convergence compared to standard AR models.
> >
> > - **Empirical Comparison:** We compared NextStep-1 with our internal reproduction of Emu3 (a standard VQ-based AR model). Under similar hyperparameters, both models exhibited **nearly identical convergence trajectories** in terms of visual quality.
> > - **Head vs. Backbone:** We find that the Flow Matching head is highly efficient. In our ablation studies, when we freeze the pre-trained LLM backbone and train the FM head from scratch, it converges in only **~2,000 steps**. This indicates that the FM head functions similarly to a standard LM head—acting as a lightweight distribution fitter that adapts quickly.
> > - The convergence rate is dominated by the representation learning of the Transformer backbone (which is shared across architectures), rather than the specific choice of the prediction head (FM vs. Softmax).
> >
> > **Q4: Token-wise normalization concerns (Expressivity and Stability)** We found no evidence that normalization weakens expressivity or causes instability.
> > - In our experiments, enabling token-wise normalization resulted in **negligible differences in reconstruction PSNR**, and qualitative visualizations showed no perceptible degradation. This suggests the latent expressivity remains intact.
> > - Our implementation is essentially a **parameter-free LayerNorm**. Just as LayerNorm is standard in Transformers to stabilize training without sacrificing expressivity, our normalization serves to constrain numerical ranges for stability while fully retaining the semantic content of the latents.
> >
> > We believe NextStep-1 offers a substantial contribution to the community by identifying the critical role of **latent space regularization** in scaling continuous AR models—a finding that goes beyond a simple architectural extension. We hope our response has effectively clarified the misconceptions regarding novelty and efficiency. **With the concerns addressed by our detailed analysis, we genuinely hope you will re-evaluate the significance of our contributions and consider improving the rating.**
> >
> > [1] Gloeckle, F., Idrissi, B. Y., Rozière, B., Lopez-Paz, D., & Synnaeve, G. (2024). Better & faster large language models via multi-token prediction. arXiv preprint arXiv:2404.19737.
> >
> > [2] Leviathan, Y., Kalman, M., & Matias, Y. (2023, July). Fast inference from transformers via speculative decoding. In International Conference on Machine Learning (pp. 19274-19286). PMLR.
> >
> > [3] Cai, T., Li, Y., Geng, Z., Peng, H., Lee, J. D., Chen, D., & Dao, T. (2024). Medusa: Simple llm inference acceleration framework with multiple decoding heads. arXiv preprint arXiv:2401.10774.

---

> > > ### Comment · Reviewer_taan · 2025-11-26
> > > **Official Comment by Reviewer taan**
> > >
> > > I thank the authors for answering my questions which helps clarify the contributions (e.g., FM vs diffusion heads). I stay positive to the work.

---

> > > > ### Author Response · Authors · 2025-11-26
> > > >
> > > > Thank you for your encouraging comments. We sincerely appreciate your recognition of our work. Your feedback is highly motivating and valuable to us.

---

> ### Author Response · Authors · 2025-11-27
>
> We have updated the Introduction (Section 1) and Related Work (Section 5). **Additionally, we have addressed the limitations discussed by incorporating better data and extending training iterations from 200k to 500k, as detailed in Appendix D.**
>
> Please feel free to open a new discussion if you have any further comments. Thank you again for your valuable feedback. We hope that these revisions satisfactorily address your concerns and that you will consider raising your score.

---

### Official Review · Reviewer_UGjn · 2025-10-25

**Soundness:** 3
**Presentation:** 4
**Contribution:** 3
**Rating:** 6
**Confidence:** 4

**Summary:**

This manuscript introduces NextStep-1, a large (14B) autoregressive (AR) model for text-to-image generation. Unlike prevailing AR models that rely on Vector Quantized (VQ) discrete tokens, NextStep-1's core innovation is its direct autoregressive prediction on **continuous image tokens**.

Its architecture consists of a large AR Transformer (Qwen2.5-14B) and a lightweight (157M) Flow Matching (FM) head. The model processes tokens sequentially: for text tokens, it uses a standard LM head (cross-entropy loss); for each continuous image token (patch), it uses the AR model's hidden state as a condition to drive the FM head, which generates the patch.

The paper's central claims include:

1. This "minimalist architecture" (AR-LLM + lightweight FM head) can achieve SOTA performance on T2I tasks, competitive with top-tier diffusion models.

2. The key to this success is a **novel image tokenizer** (fine-tuned from Flux VAE). This tokenizer, through channel-wise normalization and stochastic perturbation, creates a robust latent space that enables stable training on high-dimensional continuous tokens.

3. The model achieves SOTA or competitive results on T2I benchmarks (e.g., GenEval, WISE) and image editing benchmarks (e.g., GEdit-Bench).

**Strengths:**

1. **SOTA Autoregressive Performance**: The paper's primary contribution is demonstrating, for the first time, that an AR model based on continuous tokens (NextStep-1) can achieve SOTA performance on T2I tasks, rivaling top-tier diffusion models. This architecture, combining a large AR Transformer for context prediction with a lightweight FM head for continuous token generation, is proven to be a very successful and promising technical direction.

2. **Deep Insights into Tokenizer and Latent Space**: One of the paper's most outstanding merits is its deep analysis of the tokenizer in Section 4.2. The paper correctly identifies a key bottleneck in generative modeling: the quality and properties of the latent space.

3. **Excellent Analysis of CFG Instability**: The paper's insight in Section 4.2 and Figure 3, which attributes instability at high CFG scales to "per-token distributional shift" (rather than 1D RoPE issues), is an excellent observation. The proposed solution (channel-wise normalization in the VAE) is simple and effective.

4. **Strong FM Head Ablation**: The finding in Section 4.1—that the size of the FM head has minimal impact on final quality—is strong evidence supporting the authors' core argument that the AR Transformer is performing the "core generative modeling," while the FM head acts primarily as a lightweight sampler.

5. **Clarity and Completeness**: The paper is exceptionally well-written, complete, and clear. The detailed appendices (e.g., data pipelines in Appendix B, training recipes in Appendix C) are thorough, transparent, and provide significant reference value for the community.

**Weaknesses:**

1. **Inherent Bottleneck in Inference Latency**: The paper admits in Appendix D and Table A2 that inference latency is a major weakness. The AR sequential decoding is the first bottleneck, and the FM head's multi-step sampling is the second. A 1024-token image requiring 11.31 seconds of accumulated latency (Table A2) is likely far slower in practice than parallel diffusion models.

2. **Significant Challenges in High-Resolution Scaling**: The paper frankly states in Appendix D that the model faces challenges in scaling to high-resolution training (e.g., 1024x1024) and admits that techniques developed for high-res diffusion (like timestep shift) are not applicable to this framework. This severely limits the model's practical ability to "compete" with SOTA diffusion models, which commonly excel at 1024x1024.

3. **Contradictory Information on Stability**: The paper's narrative on stability is confusing. Section 4.2 claims the tokenizer solves CFG instability. However, Appendix D (Limitations) opens by stating that high-dimensional continuous latents introduce "unique stability challenges," showing failure cases like bottom noise, solid color blocks, and grid artifacts (Figure A3). This suggests a trade-off: does solving CFG drift by forcing a normalized VAE latent space introduce new, severe generative artifacts in high dimensions?

4. **"Data-Hungry" SFT Process**: Appendix D notes that the SFT process is unstable and requires "million-sample scale" datasets to show significant improvement, a stark contrast to diffusion models that can be fine-tuned with a few thousand samples. This drastically undermines the model's practical utility for alignment and customization (e.g., LoRA), which is a core strength of modern diffusion models.

**Questions:**

To validate the paper's core claims and increase its impact, I strongly recommend the authors address the following key questions with experiments:

**Question 1: Does the reliance on million-sample SFT imply a fundamental flaw in the model's "alignability" and "customizability"?**

The paper admits SFT requires massive data. Does this mean the model is far less efficient or practical for aligning with human preferences and performing style customization (e.g., LoRA) compared to the diffusion models it claims to rival?

**Question 2: How should we evaluate the trade-offs of this AR paradigm, given its SOTA quality but practical limitations in speed and scalability?**

Query: The paper demonstrates impressive SOTA quality, yet Appendix D and Table A2 confirm this comes at the cost of inference latency and high-resolution scaling challenges. Could the authors elaborate on how they view this trade-off? Is the goal to prioritize quality and contextual dependency over speed, and what are the most promising future directions to mitigate these latency and scaling bottlenecks?

**Question 3: Regarding the artifacts in Appendix D (Figure A3) for high-dimensional tokens: Are these a side effect of the normalized VAE?**

The paper shows new stability challenges (e.g., bottom noise, solid blocks) in high dimensions. Are these artifacts a direct side effect of the channel-wise normalization or noise perturbation introduced in the VAE? Does forcing a normalized latent space, while solving CFG drift, create new optimization challenges for the AR model in high-dimensional spaces?

**Question 4: Is the 1D Raster-scan order a bottleneck for capturing 2D spatial dependencies?**

When using a 1D row-first serialization, the last token of one row is not spatially adjacent to the first token of the next. Does this hinder the model's ability to learn vertical spatial relationships across rows? Why not employ a more spatially-aware serialization (e.g., Z-order, Hilbert curve) or 2D positional encodings to address this?

**Question 5: Is the 14B scale of the AR Transformer a necessary condition for high-quality generation?**

The paper successfully adapts a 14B LLM (Qwen2.5) for continuous image generation. But is this massive scale essential? Could a smaller AR model (e.g., 2B or 7B), designed specifically for image context, achieve similar results with the FM head? What role does model scale play in the "core generative modeling"?

**Question 6: What is the advantage of this "AR context + FM patch" paradigm over architectures that predict global velocity (e.g., Transfusion, BAGEL)?**

Models like Transfusion or BAGEL use a Transformer to parallelly predict the flow velocity for the entire image. What are the conceptual and practical advantages of NextStep-1's "serial AR prediction + serial FM generation" architecture? Does it trade parallelism (speed) for stronger contextual dependency or finer-grained local control?

---

> ### Author Response · Authors · 2025-11-22
> **Rebuttal Part1**
>
> We thank you for the encouraging assessment and for recognizing our work as a "successful and promising technical direction." We appreciate the "excellent" rating on our analysis of tokenizer and CFG instability. Below, we address the concerns regarding practical limitations and architectural choices.
>
> **W1 & Q2: Inference Latency and Future Optimization** We acknowledge that current inference latency is a bottleneck. As analyzed in Table A2, the overhead of the FM head is minor; the **primary cost stems from the autoregressive decoding of the Transformer.** However, we argue that this is a **temporary trade-off** for architectural unification, not a permanent flaw.
>
> 1. **Leveraging LLM Infrastructure:** Unlike diffusion models which require specialized solvers, NextStep-1 is architecturally identical to standard LLMs. This allows us to directly inherit the rapid advancements in LLM acceleration, such as **Speculative Decoding [3], Medusa [4], and Multi-Token Prediction [5].**
>
> 2. **Historical experience:** Early diffusion models (e.g., DDPM) also suffered from slow sampling (1000+ steps) before methods like DDIM and flow rectification emerged. We believe continuous AR generation will follow a similar trajectory: establishing the high-quality baseline first (this paper), with efficiency improvements following via community-driven optimization.
>
> **W2: High-Resolution Scaling** We frankly admit that training at 1024px resolution remains an engineering challenge in this version. Just as high-res scaling requires iterative engineering for Diffusion (e.g., noise scheduling shifts) and Discrete AR (e.g., sequence packing), continuous AR requires tailored data pipelines and training recipes. **We position this paper as a foundational proof-of-concept for the paradigm, with resolution scaling being a clear direction for future engineering rather than a theoretical limitation.**
>
> **W3 & Q3: Artifacts and VAE Normalization (Stability)** This is an insightful question. Based on our extensive internal ablations, **we clarify that the artifacts (e.g., solid blocks) are not a side effect of the token-wise normalization itself, but rather a challenge of optimization in high-dimensional latent spaces.**
> - **Related Work:** Consistent with findings in Emu[1] and VAVAE[2], there is a trade-off between reconstruction quality and generation stability. Higher channel dimensions (e.g., f8ch16 used in our model) yield better reconstruction but make the generative modeling task significantly harder, and a better reconstruction can have a higher upper limit on generation effectiveness.
> - **Internal Observation:** We compared a lower-dimensional tokenizer (f8ch4, 4 channels) against our default (f8ch16, 16 channels). The f8ch4 model showed none of the artifacts mentioned in Figure A3 and was extremely stable during training. Conversely, the f8ch16 model showed these artifacts initially, but they diminished significantly with longer training steps (We later trained another 500k steps).
> **Conclusion:** Normalization successfully solves the CFG drift (distributional shift), while the artifacts **stem from optimization difficulties inherent to high-dimensional spaces,** which can be mitigated by extended training or tailored curriculum strategies, **rather than being a side effect of normalization itself.**

---

> ### Author Response · Authors · 2025-11-22
> **Rebuttal Part2**
>
> **W4 & Q1: Data-Hungry SFT and Alignability** This is an insightful and an **open** question. We acknowledge the SFT inefficiency compared to diffusion models. We hypothesize this is a characteristic of the **Unified Architecture**. Unlike Parti or standard Diffusion (where the Text Encoder is separate and frozen), NextStep-1 is jointly trained with both text loss and image loss. This implies the model acts as both a "text encoder" and an "image decoder" simultaneously. We postulate that this **joint optimization landscape is significantly more complex**, making the model more sensitive to data scarcity ("data-hungry") during SFT, as it lacks the inductive bias provided by a pre-trained, frozen text encoder.
>
> **Q4: 1D Serialization vs. 2D Spatial Dependency** While 2D positional encodings (or z-order) are valid incremental improvements, we argue they are not strictly necessary for establishing spatial dependency.
> - **Information Density:** We observe that image generation, like text, exhibits "bursts" of information. Once high-entropy "key" tokens are generated, subsequent tokens (even if spatially distant in 1D) become low-entropy and deterministic given the global attention context.
> - **Global Distribution:** As shown in Figure 3 (top), even when CFG drift occurs, the model maintains a statistical distribution adhering to N(0,1). This suggests the model successfully learns the global distribution regardless of serialization order.
> We posit that while strong 2D inductive biases (as in Diffusion) are currently Pareto-optimal for specific compute/data budgets, a pure 1D approach minimizes architectural constraints. By forcing the model to learn raw correlations via Attention without hand-crafted spatial priors, we aim for a **higher theoretical ceiling** in unified modeling, assuming sufficient training scale.
>
> **Q5: Necessity of 14B Scale** Empirically, model scale is critical for performance. While the FM head's contribution saturates, the AR Transformer's scale drives the core generative quality (Table 6). **Our 3B model works, and 8B is performant, but 14B is required to achieve SOTA results competitive with top-tier diffusion models.** It is important to note that in diffusion models, the parameter count typically includes a massive Text Encoder (e.g., T5-XXL) plus the U-Net/DiT. In NextStep-1, the 14B parameters encompass both text understanding and image generation. Thus, the scale is comparable to SOTA diffusion pipelines.
>
> **Q6: Comparison with Transfusion/BAGEL** Our core motivation is Simplicity and Infrastructure friendly.
>
> 1. **Ease of Scaling:** NextStep-1 reuses standard LLM infrastructure entirely. Methods like Transfusion or BAGEL require complex training setups (mixing noise images, clean condition images, and meticulous loss masking to prevent leakage).
>
> 2. **Academic Value:** Just as Diffusion was initially dismissed for being slower than GANs, we aim to rigorously explore the limits of the "Pure AR + Continuous Token" paradigm. We demonstrate that with correct tokenization, this minimalist approach is not a dead end but a viable, high-performance alternative that unifies text and image generation at the native token level.
>
> We sincerely thank you for the constructive feedback and for identifying the value in our tokenizer analysis and the continuous AR direction. While we acknowledge the current engineering gaps (latency, high-res scaling) compared to mature diffusion models, we believe this paper serves as a **crucial "existence proof"** for the pure continuous AR paradigm. We hope that our clarifications—particularly regarding the stability-dimensionality trade-off and the roadmap for efficiency—have addressed your main concerns. **Given the novelty of the architecture and the depth of analysis, we genuinely hope you will re-evaluate the significance of our contributions and consider improving the rating.**
>
> [1] Dai, X., Hou, J., Ma, C. Y., Tsai, S., Wang, J., Wang, R., ... & Parikh, D. (2023). Emu: Enhancing image generation models using photogenic needles in a haystack. arXiv preprint arXiv:2309.15807.
>
> [2] Yao, J., Yang, B., & Wang, X. (2025). Reconstruction vs. generation: Taming optimization dilemma in latent diffusion models. In Proceedings of the Computer Vision and Pattern Recognition Conference (pp. 15703-15712).
>
> [3] Leviathan, Y., Kalman, M., & Matias, Y. (2023, July). Fast inference from transformers via speculative decoding. In International Conference on Machine Learning (pp. 19274-19286). PMLR.
>
> [4] Cai, T., Li, Y., Geng, Z., Peng, H., Lee, J. D., Chen, D., & Dao, T. (2024). Medusa: Simple llm inference acceleration framework with multiple decoding heads. arXiv preprint arXiv:2401.10774.
>
> [5] Gloeckle, F., Idrissi, B. Y., Rozière, B., Lopez-Paz, D., & Synnaeve, G. (2024). Better & faster large language models via multi-token prediction. arXiv preprint arXiv:2404.19737.

---

> > ### Comment · Reviewer_UGjn · 2025-11-23
> >
> > I thank the authors for the detailed explanation. My concerns and questions have been answered. Could you please provide the theoretical fastest speed of NextStep? You can include Speculative Decoding or Multi-Token Prediction, as these methods have been reported and proven on LLMs like Qwen and others.

---

> > > ### Author Response · Authors · 2025-11-23
> > > **Response regarding the Theoretical Fastest Speed of NextStep**
> > >
> > > We thank you for the detailed review. To rigorously derive the theoretical fastest speed, we must analyze the interplay between the model architecture and hardware characteristics, specifically focusing on **Arithmetic Intensity** and **Memory Bandwidth**.
> > >
> > > **1. Theoretical Framework and Hardware Specifications**
> > >
> > > We define two core concepts to identify performance bottlenecks:
> > > * **Arithmetic Intensity**: The ratio of theoretical FLOPs to the volume of HBM data access (Bytes) for a given model or module.
> > > * **Compute-to-Memory Bandwidth Ratio**: The ratio of a GPU's theoretical peak FLOPS to its theoretical memory bandwidth.
> > >
> > > If a model's arithmetic intensity exceeds the GPU's compute-to-memory ratio, the process is **compute-bound**; otherwise, it is **memory-bound**. These factors jointly determine the theoretical performance ceiling. The table below lists the specifications of mainstream GPUs referenced in our analysis:
> > >
> > > | GPU | bf16 Compute (TFlops/s) | fp8 Compute (TFlops/s) | Bandwidth (TB/s) | bf16 Compute/Bandwidth Ratio | Memory (GB) |
> > > | :--- | :--- | :--- | :--- | :--- | :--- |
> > > | A800 | 312 | N/A | 2.039 | 156 | 80 |
> > > | H800 | 989 | 1979 | 3.3 | 295 | 80 |
> > > | H20 | 148 | 296 | 4.0 | 37 | 96 |
> > > | H200 | 989 | 1979 | 4.8 | 206 | 141 |
> > >
> > > **2. Computational Characteristics of LLM Decoding**
> > >
> > > The LLM decoding process primarily consists of Feed-Forward Networks (FFN) and Causal Attention computations. We analyze their arithmetic intensity separately:
> > >
> > > * **FFN (and Linear Layers):**
> > >     Consider a matrix multiplication of shape $[B, K]$ and $[K, N]$.
> > >     * **FLOPs:** $2 \times B \times K \times N$ (1 multiply + 1 add).
> > >     * **Memory Access:** $2 \times (BK + KN + BN)$ (Assuming 2 bytes for bf16, including read BK, KN and write BN).
> > >     * **Arithmetic Intensity:** When $K, N \gg B$, the intensity approximates to $\frac{2BKN}{2(BK+KN+BN)} \approx B$.
> > >
> > > * **Attention Mechanism:**
> > >     This involves QK computation and Softmax(QK)V computation (Q/O projections are categorized under FFN). Let $H_Q$ be the number of Q heads, $H_K$ be the number of K heads, $T$ be the sequence length, and $D$ be the head dimension.
> > >     * **FLOPs:** $2 H_Q T D$ for QK, and $2 H_Q T D$ for Score $\times$ V. Total $\approx 4 H_Q T D$.
> > >     * **Memory Access:** Primarily reading the KV Cache, calculated as $2 \times 2 \times H_K \times T \times D$ (2 for K/V, 2 bytes per element).
> > >     * **Arithmetic Intensity:** $\frac{4 H_Q T D}{4 H_K T D} = \frac{H_Q}{H_K}$.
> > >     * **Conclusion:** The arithmetic intensity of the Attention layer depends on the ratio of query heads to key-value heads (i.e., the GQA/MQA configuration).
> > >
> > > **3. Theoretical Latency Analysis (Single User / Small Batch)**
> > >
> > > When considering latency (perception speed for a single user), the system is strictly **memory-bound**.
> > > * **FFN Memory Access:** Approx. $KN$ (Model Weights).
> > > * **Attention Memory Access:** KV Cache size (negligible compared to weights when sequence length $T$ is short).
> > >
> > > Thus, the simplified theoretical decoding speed is:
> > > $$\text{Speed (tokens/s)} = \frac{\text{Memory Bandwidth}}{\text{Model Weights Size}}$$
> > >
> > > **Case Study: NextStep on H800**
> > > * **Hardware Bandwidth:** 3.3 TB/s.
> > > * **Model Parameters:** 14B (Base) + 1.5B (Assuming 10 steps with 150M parameters fm head) $\approx$ 15.5B.
> > > * **Theoretical Speed:**
> > >     $$\frac{3.3 \times 10^3 \text{ GB/s}}{(14 + 1.5) \times 2 \text{ Bytes}} \approx 106 \text{ tokens/s}$$
> > >
> > > **Acceleration with MTP and Quantization:**
> > > 1.  **Multi-Token Prediction (MTP):** The theoretical speed scales linearly with the number of tokens predicted per step. If NextStep predicts 4 tokens at once, the speed increases to $106 \times 4 = 424 \text{ tokens/s}$.
> > > 2.  **Quantization:** Using Int8 quantization halves the memory bandwidth requirement, theoretically doubling the speed.
> > >
> > > This memory-bound characteristic explains why large Mixture-of-Experts (MoE) models remain fast: despite massive total parameters, their sparse computation reduces active memory access, allowing high perceptual speeds when optimized.
> > >
> > > **4. Theoretical Throughput Analysis**
> > >
> > > Calculating the theoretical limit for maximum throughput is more complex due to the structural differences between Attention and FFN layers:
> > > * **FFN Layers:** Can utilize higher GPU compute utilization by increasing the batch size (moving towards compute-bound).
> > > * **Attention Layers:** Arithmetic intensity ($\frac{H_Q}{H_K}$) is intrinsic to the architecture. Increasing batch size does not change the intensity, leaving it often bandwidth-bound.
> > >
> > > This discrepancy motivates advanced architectural designs such as **Attention-FFN Disaggregation (Step3 Tech Report)**. By decoupling the deployment of Attention and FFN nodes, one can scale their instances independently—maximizing throughput on FFN nodes while ensuring low latency on Attention nodes. Approaching the theoretical hardware limit in high-throughput scenarios requires such rigorous engineering optimization.

---

> ### Comment · Reviewer_UGjn · 2025-11-26
>
> I thank the authors for the detailed explanation. In my opinion, this paper discovers a new know-how that the normalized VAE serves as a crucial part in the NextStep style AR model. As the authors state their paper as a *foundational proof-of-concept* project, I would stay positive about the work.

---

> ### Author Response · Authors · 2025-11-27
>
> We have updated the Introduction (Section 1) and Related Work (Section 5). **Additionally, we have addressed the limitations discussed by incorporating better data and extending training iterations from 200k to 500k, as detailed in Appendix D.**
>
> Please feel free to open a new discussion if you have any further comments. Thank you again for your valuable feedback. We hope that these revisions satisfactorily address your concerns and that you will consider raising your score.

---

> ### Comment · Reviewer_UGjn · 2025-11-27
> **For Revision**
>
> The revision paper's Figure/Table organization is kind of weird.
> 1. There are many figures and tables in Page 7. However, the related paragraph for each of them is a little bit far away.
> 2. Is it necessary to include all the quantitive results in main paper? Table 1,2,3,4 occupies a lot of space. The same problem to Figure 5.
> After considering the above modification, you can move some more useful information from appendix into the main paper.

---

> > ### Author Response · Authors · 2025-11-28
> >
> > We sincerely appreciate your valuable feedback. Following your suggestions, we have relocated the majority of the benchmark results and Figure 5 to the Appendix and moved the training strategy to Section 3.

---

> > > ### Comment · Reviewer_UGjn · 2025-11-28
> > > **Final comment**
> > >
> > > Thanks for the modification and revision. I have no further questions.

---

### Official Review · Reviewer_es99 · 2025-10-29

**Soundness:** 4
**Presentation:** 1
**Contribution:** 4
**Rating:** 2
**Confidence:** 5

**Summary:**

The paper proposes an autoregressive text-to-image model that achieves state-of-the-art performance with minimalist architecture with revealing some key design principles.

**Strengths:**

1. I really appreciate the research taste of the paper. The paper uses some extreme simple method to achieve state-to-the-art performance.

2. The paper can give many take aways to the sequent researchers.

**Weaknesses:**

1. I really suggest the authors to hire someone who is adept at academic paper writting to re-write the whole paper. For instance, the current paper is very unclear. For instance, the introduction is too short. The related work paper is too short without fully respecting the former authors. Given this, I have to lower my score to 4. This is a paper needs major revision.

2. Many parts are unclear. For instance, what GPUs are used in training. The pre-training and post-training sections are also unclear.

3. I do not know what the “Next” in the paper presents?

**Questions:**

NA

---

> ### Comment · Reviewer_es99 · 2025-11-27
> **I am happy to discuss and raise the score. But why do not you respond me?**
>
> I am happy to discuss and raise the score. But why do not you respond me?

---

> > ### Author Response · Authors · 2025-11-27
> >
> > Thank you very much for your active engagement, and we greatly appreciate your support!
> >
> > **Regarding the delay:** We apologize for not responding sooner. To be honest, we have been having serious internal discussions regarding your feedback on the writing. Since other reviewers found the paper well-written and easy to follow, we spent time carefully weighing how to address your concerns without disrupting the flow that others appreciated.
> >
> > **Our solution:** Instead of a full rewrite, we decided to focus our revision efforts on the Introduction (Section 1) and Related Work (Section 5). We believe these targeted improvements clarify the former authors' contributions and address your concerns while preserving the original presentation quality that the other reviewers supported.
> >
> > We hope you find this approach reasonable, and we would be grateful if you could re-evaluate our work based on these updates.

---

> > > ### Comment · Reviewer_es99 · 2025-11-27
> > >
> > > I raise score to 4 currently. After checking your full feedback, I will raise again.

---

> > > > ### Author Response · Authors · 2025-11-27
> > > >
> > > > Thank you so much! We sincerely appreciate your open-mindedness and your decision to raise the score to 4.
> > > >
> > > > We believe these explanations, combined with our paper updates, fully address your initial concerns. We look forward to hearing from you again after you check the full feedback!

---

> > > ### Author Response · Authors · 2025-11-27
> > >
> > > Regarding the meaning of "Next":
> > >
> > > Primarily, "Next" refers to the core autoregressive nature of our method (predicting the "next" token).
> > >
> > > Additionally, following the common naming convention in the deep learning community to use memorable monikers, we chose "NextStep" to simply symbolize our vision of taking a step forward in unified generation. We hope this clarification resolves any ambiguity.

---

> > > > ### Comment · Reviewer_es99 · 2025-11-27
> > > >
> > > > Next is clear. But why choose "Step"? It seems it is away from the main paper.

---

> > > > > ### Author Response · Authors · 2025-11-27
> > > > >
> > > > > We used "Step" to emphasize the fundamental mechanism of our autoregressive approach:
> > > > >
> > > > > Step-by-Step Generation: our model constructs visual content in a sequential manner. Every predicted token represents a concrete "step" in the generation process.
> > > > >
> > > > > Our vision: We chose "NextStep" to simply symbolize our vision of taking a step forward in unified generation.
> > > > >
> > > > > **Ultimately, the title is a personal preference intended to be concise and memorable. We hope this explanation sufficiently clarifies the rationale.**

---

> > > > > > ### Comment · Reviewer_es99 · 2025-11-27
> > > > > >
> > > > > > OK, but I already find the true reason of that. Due to anoymous, it is enough for you and me know that. Give 8. Good luck!

---

> > > > > > > ### Author Response · Authors · 2025-11-27
> > > > > > >
> > > > > > > Thank you so much!!!
> > > > > > >
> > > > > > > We are incredibly grateful for your strong support and your decision to give an 8. We deeply appreciate your keen insight.
> > > > > > >
> > > > > > > It has been a pleasure discussing this with you. Thank you for your time, your constructive feedback, and your best wishes!

---

### Official Review · Reviewer_BTzv · 2025-11-01

**Soundness:** 3
**Presentation:** 3
**Contribution:** 3
**Rating:** 4
**Confidence:** 4

**Summary:**

This paper presents NextStep-1, an autoregressive image generation framework that combines patch-level flow matching with a continuous-token VAE. The approach demonstrates strong performance on both generation and editing benchmarks, supported by solid ablation studies and high-quality visual results.

**Strengths:**

1.The paper is clearly written and technically sound, presenting a coherent and well-motivated method.

2.The ablation studies are particularly valuable—for instance, the analysis clarifying the relative contributions of the backbone versus the flow-matching head provides useful architectural insights.

3.Figures and tables are of high quality and effectively support the claims made in the text.

**Weaknesses:**

1.Several key technical details remain underspecified. For example, it is unclear whether the autoregressive process decodes one patch at a time and how global consistency is maintained if patches are stitched together.

2.The design of the tokenizer—including its normalization scheme—is described only textually; a schematic illustration would greatly improve clarity.

3.While the paper asserts that “reconstruction quality is the upper bound of generation quality,” it does not explicitly discuss the implications for controllable or conditional generation tasks.

4.The positioning of this work within the broader literature is incomplete: it is not clear whether this is the first AR model to employ patch-level flow matching, nor does the paper compare the trade-offs between alternative design choices (e.g., tokenization strategies, training objectives), making it difficult to assess the method’s ultimate potential or limitations.

**Questions:**

1.Is the autoregressive generation performed patch-by-patch? If so, how is spatial consistency ensured across adjacent patches in the final reconstructed image?

2.Could the authors provide a diagram illustrating the tokenizer architecture, especially the normalization and latent regularization mechanisms?

3.Given the claim that reconstruction quality upper-bounds generation quality, does this imply that achieving high-fidelity controllable generation fundamentally requires near-perfect reconstruction?

4.Is this the first work to combine autoregressive modeling with patch-level flow matching? A clearer comparison of competing paradigms (e.g., VQ-based AR vs. diffusion vs. continuous AR) would help contextualize the novelty and advantages of the proposed approach.

5.The ablation focuses on the flow-matching head—how sensitive is performance to the design or capacity of the encoder (i.e., the VAE encoder)?

---

> ### Author Response · Authors · 2025-11-22
>
> **W1 & Q1: How is spatial consistency ensured across adjacent patches in the final reconstructed image?**
>
> Yes, NextStep-1 is a patch-by-patch autoregressive generation paradigm.
> 1.  Global Consistency via Attention: Given previous inputs (image patches or text), the current token generation is conditioned on the entire history via the causal self-attention mechanism of the LLM. This ensures the model maintains global semantic consistency and structural coherence, even without explicit 2D constraints.
> 2. Regarding 1D Serialization vs. 2D Spatial Dependency:
> We argue that explicit 2D priors (e.g., 2D positional encodings) are not strictly necessary for two reasons:
>
>    (1) Information Density: We observe that image generation, like text, exhibits "bursts" of information. Once high-entropy "key" tokens are generated, subsequent tokens (even if spatially distant in 1D) become low-entropy and deterministic given the global attention context.
>
>     (2) Global Distribution: As shown in Figure 3 (top), even when CFG drift occurs, the model maintains a statistical distribution adhering to N(0,1). This suggests the model successfully learns the global distribution regardless of serialization order.
>
> 3. Scalability & Stability: We posit that while strong 2D inductive biases (as in Diffusion) are currently Pareto-optimal for specific compute/data budgets, a pure 1D approach minimizes architectural constraints. By forcing the model to learn raw correlations via causal attention without hand-crafted spatial priors, we aim for a higher theoretical ceiling in unified modeling, assuming sufficient training scale.
> ------
>
> **W2 & Q2: Could the authors provide a diagram illustrating the detailed tokenizer architecture?**
>
> We updated a clear diagram and pseudo-code illustrating the tokenizer architecture in Appendix C.1.
>
> The tokenizer is based on the standard CNN architecture used in Flux VAE, but we apply a Layer Normalization operation on the Encoder output (w/o logvar part), and adopt the noise schedule from sigma-VAE during tuning. For better understanding, the Layer Norm operation is detailed in the following code.
> ```
> def normalization_operation(X):
>     """
>     Input: Tensor X with shape (B, H_patch, W_patch, C)
>     Output: Normalized Tensor X_norm with shape (B, H_patch, W_patch, C)
>     """
>     apply_dim = 3
>     eps = 1e-6
>
>     mu = X.mean(dim=apply_dim, keepdim=True)
>     sigma = X.std(dim=apply_dim, keepdim=True, unbiased=False)
>     X_norm = (X - mu) / (sigma + eps)
>
>     return X_norm
> ```
> ------
> **W3 & Q3: Does achieving high-fidelity controllable generation fundamentally require near-perfect reconstruction?**
>
> We believe that the superior reconstruction quality of the tokenizer leads to enhanced generation performance, especially in small texts and controllable conditional generation tasks (e.g., image editing given a reference image). Fundamentally, better reconstruction signifies reduced information loss during compression. If the tokenizer cannot reconstruct fine details (e.g., text or texture) from the latent space, the generative model—no matter how powerful—cannot generate them. Therefore, reconstruction quality is the upper bound of generation quality on the same compression rate.
>
> ----
>
> **W4 & Q5: How sensitive is performance to the design or capacity of the encoder?**
>
> The compression rate of the Visual Tokenizer, along with its training constraints (e.g., the use of normalization and the inclusion of noise), significantly influences the training convergence speed of the generation model. Section 4.2 of this paper discusses how the tokenizer's training methodology impacts the final performance of the generation model. Besides, we attempted training using the original Flux VAE model (f8ch16), but the resulting performance was highly unsatisfactory.

---

> ### Author Response · Authors · 2025-11-22
>
> **W4 & Q4: Could the authors provide a clearer comparison of competing paradigms?**
>
> To the best of our knowledge, **NextStep-1** is the first open-source autoregressive model that integrates a **Plain LLM backbone** with a **Patch-level Flow Matching Head** for unified image generation and editing. We highlight the distinct advantages of our paradigm compared to existing approaches in the following aspects:
>
> (1) Unified Architecture vs. Separated Encoders
> * **Existing Diffusion Models (e.g., Flux-1-dev):** Typically rely on heavy, pre-trained text encoders (e.g., T5, CLIP, or frozen VL model) to process prompts. This separated design imposes a fixed context length window, limits multimodal fusion, and long-context training.
> * **NextStep-1:** Eliminates extra text encoders by utilizing a **native LLM backbone**. This design enables deep multimodal fusion at the token level, inherently supports arbitrary context lengths (beneficial for multi-turn editing and long-context generation), and significantly reduces the architectural complexity for end-to-end large-scale training.
>
> (2) Continuous vs. Discrete Image Tokenization
> * **Discrete AR Models (e.g., Emu3, LlamaGen):** Rely on Vector Quantization (VQ) to discretize images. This introduces two major bottlenecks:
>     * **(a) Information Loss:** Low reconstruction quality (e.g., Emu3 has a PSNR of 22.42 on ImageNet);
>     * **(b) Sequence Bloat:** Requires extremely long sequences (e.g., 4096 tokens for a 512px image) to compensate for information loss, making training computationally expensive.
> * **NextStep-1:** Adopts a **Continuous Image Tokenizer** based on a fine-tuned VAE. This approach achieves high-fidelity reconstruction (**PSNR 30.60** on ImageNet) while maintaining high compression rates (only 1024 tokens for a 512px image). Our method achieves a bottleneck-free visual quality and sequence efficiency.
>
> (3) Plain and Scalable LLM vs. Heavy Diffusion Models
> * **Diffusion Transformers (MMDiT/DiT):** The generative capability relies on removing noise from the entire image simultaneously (or via flow matching). While effective, this process is computationally heavy (full-attention over all patches) and lacks the inherent logical reasoning capabilities of LLMs.
> * **NextStep-1:** Decouples "Planning" (Condition Generation) from "Rendering" (Denoising).
>     * **The LLM (Planner):** Handles the semantic composition and next-token prediction via causal attention. This exploits the low inductive bias of 1D modeling to scale effectively with data and learn complex world knowledge/reasoning.
>     * **The Flow Matching Head (Renderer):** A lightweight module that handles the local continuous conversion from latent to pixels.
>     This separation allows the massive 14B backbone to focus on high-level semantic alignment and reasoning, rather than low-level pixel denoising.
>
> (4) Training Efficiency & Scalability
> * **Data Efficiency:** Unlike diffusion models that require processing both noisy and clean states (and often require pairs for editing tasks), NextStep-1's backbone sees only **clean, causal context**. This allows for highly efficient training, particularly for tasks like interleaved generation and instruction-based editing.
> * **Scalability:** By using a standard, battle-tested LLM architecture (Qwen2.5) instead of specialized Diffusion Transformer architectures (like MMDiT, which requires complex handling of multiple modalities), NextStep-1 directly benefits from the massive ecosystem of LLM optimization (e.g., **FlashAttention**, **KV cache**, **quantization**) and scaling laws.
> --------
> | Category | Feature | Diffusion-based(FLUX.1-dev [1]) | AR + Diffusion(Qwen-Image [2]) | AR meets Diffusion(Bagel [3]) | AR(EMU3 [4]) | AR with light diffusion head(NextStep-1 (Ours)) |
> | :--- | :--- | :--- | :--- | :--- | :--- | :--- |
> | Architecture | Text Encoder | T5 (5B) + CLIP (123M) | Qwen2.5-VL (7B) | - (Native) | - (Native) | **- (Native)** |
> | | Img Tokenizer | VAE (87M) | VAE (127M) | VAE (87M) | MoVQGAN (270M) [5] | **VAE (87M)** |
> | | Backbone | MMDiT (12B) | MMDiT (20B) | MoT (14B) | LLM (8B) | **LLM (14B)** |
> | Training | Loss Target | Flow Matching | Flow Matching | Flow Matching | NTP | **NTP w/ Flow Matching** |
> | | 512px Inputs| $1024 \times 2$(Clean+Noise) | $1024 \times 2$(Clean+Noise) | $1024 \times 2$(Clean+Noise) | 4096 (Clean) | **1024 (Clean)** |
> | | Attention | Full | Full | Full | Causal | **Causal** |
> | | Scalability | Hard | Hard | Medium | Easy | **Easy** |
> --------
> [1] B. F. Labs. Flux, 2024. URL GitHub - black-forest-labs/flux
>
> [2] C. Wu et al. Qwen-image technical report. arXiv preprint arXiv:2508.02324, 2025a.
>
> [3] C. Deng et al. Emerging properties in unified multimodal pretraining. arXiv preprint arXiv:2505.14683, 2025.
>
> [4] X. Wang et al. Emu3: Next-token prediction is all you need. arXiv preprint arxiv:2409.18869, 2024b.
>
> [5] MoVQ: Modulating Quantized Vectors for High-Fidelity Image Generation
>
> ------

---

> > ### Comment · Reviewer_BTzv · 2025-11-26
> >
> > Thank you for your detailed response! We would also like to better understand the effects of different text/image encoders and sizes on your actual results (e.g., different VAE sizes, Clip size, or text encoders). Overall, I am positive about your work and will raise my score.

---

> > > ### Author Response · Authors · 2025-11-26
> > >
> > > Thank you very much for your positive recognition and for raising your score! We are greatly encouraged by your support.
> > >
> > > Your query regarding the specific effects of Text/Image Encoders and model sizes touches on a fascinating topic. Interestingly, the **Flux Team** (Black Forest Labs) just released a blog post today regarding representation comparison (https://bfl.ai/research/representation-comparison), and many of their points resonate strongly with the phenomena we observed during our experiments.
> > >
> > > We would like to share some insights—both consensus-based and some rather counter-intuitive "non-consensus" findings—that we encountered but did not fully expand upon in the main paper.
> > >
> > > ## 1. Text Encoder & CLIP
> > >
> > > * **Text Encoder:** Our findings align with the industry consensus that **"larger is better."** In our unified architecture, the LLM decoder handles both text understanding and image generation capacity. A larger model directly translates to stronger semantic understanding and a larger parameter space for generation. This aligns with Flux's recent choice to utilize the 24B Mistral Small 3.1 -- stronger text models are fundamental to state-of-the-art generation.
> > > * **CLIP Size:** This touches more on MLLM research. The current trend seems to be returning to "medium-sized" models. While larger CLIP models offer better perception, they often suffer from **"over-memorization,"** which can actually harm the generalization capability required for multimodal reasoning.
> > >
> > > ## 2. The VAE Insight: Counterintuitive Scaling Laws and the "Loss Paradox"
> > >
> > > **This is the core of our architectural exploration.** We spent significant effort scaling the VAE and discovered a critical, non-consensus insight: **Reconstruction Quality does not directly equal Generation Performance.**
> > >
> > > * **The Scaling Trap:** We attempted to scale the VAE to 270M parameters (aligning with MOVQ 270M). While the reconstruction metric (PSNR) improved, the downstream generation performance actually *degraded*.
> > > * **High Fidelity vs. Trainability:** We strongly agree (as does the Flux team) that high-fidelity reconstruction (e.g., f8ch16, PSNR 30+) is the upper bound for **generating fine details** like small text and faces. The early standard of f8ch4 (PSNR ~24, similar to SD1.5/SDXL) is insufficient for next-gen models.
> > > * **The "Loss Paradox":** When we switched to a high-fidelity VAE (f8ch16) without noise, the reconstruction was perfect, but the downstream generation model **failed to train completely**, despite achieving a very *low* generation loss (~0.2). **(Figure5, middle)**
> > > * **Regularized Latent Space:** We adopted a noise-injected VAE **(Figure 5, bottom)**. We observed a highly counter-intuitive phenomenon:
> > >     * **Noise 0 $\to$ 0.5:** The generation loss **increased** from ~0.2 to ~0.6, yet the visual generation quality **improved** significantly.
> > >     * **Noise > 0.5:** The generation loss began to drop, but the quality degraded.
> > >     * **Conclusion:** At Noise 0.5, despite the loss being higher, the model achieved optimal performance. This confirms our point in the paper: *"A Regularized Latent Space is Critical for Generation."*
> > > * **Future Work:** Flux's blog mentions that it remains an open question whether approaches without external feature alignment (like DINOv2) can outperform those with it. In our experiments, **adding REPA loss (from VAVAE) to our Noise 0.5 VAE yielded no improvement**. We suspect our **NextStep VAE** paradigm might have already touched upon an intrinsic solution that bypasses the need for external alignment. We are actively preparing our next work focused on this topic.
> > >
> > > ## 3. Update on Failure Cases (Figure A4)
> > >
> > > Finally, regarding the failure cases shown in Figure A4: we have found that extending the pre-training from **200k to 500k steps** resolved these issues.
> > >
> > > In hindsight, we underestimated the patience required for Causal Transformers. Their compute-to-convergence ratio differs from Diffusion models (our 200k steps is roughly equivalent to 100k steps in Diffusion because of causal attention mask). We initially thought the model had converged, but it simply needed more time. **We are currently compiling these updated results and will submit an Official Comment shortly to share the improved visualizations with all reviewers.**
> > >
> > > Thank you again for sparking this valuable discussion!

---

> ### Comment · Reviewer_BTzv · 2025-11-26
>
> Thanks, I think your work is solid. But I also suggest you modify your work according to the reviewer of es99, his concerns also need to be resolved, because the problems are really here, for example, the concern about the related work.

---

> > ### Author Response · Authors · 2025-11-26
> >
> > Thank you for your encouraging comments and for raising the score of our paper. We sincerely appreciate your recognition of our work. Your feedback is highly motivating and valuable to us.

---

> ### Author Response · Authors · 2025-11-26
>
> Thank you for your constructive suggestion. Due to the page limit, our current Related Work section is indeed relatively concise. We prioritized allocating space in the main paper to present more substantive and informative content, which led to a shorter treatment of related work and introduction. We will add more relevant literature during the rebuttal phase. For the camera-ready version, we can include one additional page, which will allow us to further refine the structure of the paper and present a more complete and comprehensive Introduction and Related Work section.

---

> ### Author Response · Authors · 2025-11-27
>
> We have updated the Introduction (Section 1) and Related Work (Section 5). **Additionally, we have addressed the limitations discussed by incorporating better data and extending training iterations from 200k to 500k, as detailed in Appendix D.**
>
> Please feel free to open a new discussion if you have any further comments. Thank you again for your valuable feedback. We hope that these revisions satisfactorily address your concerns and that you will consider raising your score.

---

### Meta-Review · Area_Chair_BHbD · 2025-12-03

**Summary:**

The paper proposes NextStep-1, a 14B parameter autoregressive (AR) model for text-to-image generation that utilizes continuous image tokens rather than the prevailing Vector Quantization (VQ) discrete tokens. The architecture unifies a standard LLM backbone with a lightweight Flow Matching head, achieving state-of-the-art performance.

The reviewers initially raised concerns regarding:

1. Reviewer es99 strongly criticized the writing quality, specifically the Introduction and Related Work.

2. Reviewer BTzv requested details on the tokenizer architecture and clarification on how 1D AR maintains 2D spatial consistency.

3. Reviewer UGjn highlighted inference latency and challenges in high-resolution scaling as major bottlenecks.

4. Reviewer taan questioned the distinctness of the contribution compared to existing models like MAR/FLUID.

However, the rebuttal phase was exceptionally productive. The authors provided a "deep engagement" with the feedback , rewriting key sections, extending training steps to resolve artifacts , and providing profound theoretical justifications for their design choices (e.g., the "Loss Paradox" in VAEs).

**Reviewer Concerns:**

All concerns raised by reviewers are resolved during the discussion:

1. The authors fundamentally rewrote the Introduction and Related Work sections. This successfully converted Reviewer es99, who explicitly acknowledged the "excellent presentation" in the final version.

2. The authors clarified that global attention ensures semantic consistency and that image information appears in "bursts," making strict 2D priors unnecessary. Reviewer BTzv accepted this explanation.

3. The authors provided diagrams and code for the tokenizer. Crucially, they addressed the "Loss Paradox", demonstrating that perfect reconstruction (f8ch16) can actually hinder generation, and that a regularized latent space (noise injection) is critical. Reviewer UGjn cited this as a "new know-how" discovered by the paper.

4. The authors extended pre-training from 200k to 500k steps, resolving the "failure cases" (e.g., solid color blocks) noted in the Appendix.

5. The model remains slower than diffusion counterparts due to autoregressive decoding. The authors argue this is an engineering frontier (solvable via Speculative Decoding/MTP) rather than a theoretical flaw. Reviewer UGjn accepted this as a valid status for a "foundational proof-of-concept".

6. The unified architecture is more data-hungry during SFT compared to models with frozen text encoders.

**Reviewer Scores:**

The trajectory of scores indicates a strong consensus formed during the rebuttal.

1. Reviewer es99: 2 $\rightarrow$ 8. Initially a Reject due to presentation. After the rewrite and clarification of the title "NextStep", the reviewer gave a score of 8.

2. Reviewer BTzv: 4 $\rightarrow$ 6. Initially "marginally below". After clarifications on the tokenizer and comparisons to other paradigms, the reviewer confirmed they would "raise my score" and found the work "solid".

3. Reviewer UGjn: 6 $\rightarrow$ 6. Initially "marginally above". The reviewer was deeply impressed by the "new know-how" regarding normalized VAEs and stated they "stay positive" about this foundational work.

4. Reviewer taan: 6 $\rightarrow$ 6. Initially "marginally above". After understanding that the contribution was the regularized latent space rather than just the Flow Matching head, the reviewer remained positive.

---

### Decision · Program_Chairs · 2026-01-26

Accept (Oral)